# Tisp40 prevents cardiac ischemia/reperfusion injury through the hexosamine biosynthetic pathway in male mice

Xin Zhang[1,4], Can Hu[2,4], Zhen-Guo Ma[2], Min Hu[2], Xiao-Pin Yuan[1], Yu-Pei Yuan[2], Sha-Sha Wang[3], Chun-Yan Kong[2], Teng Teng[2] & Qi-Zhu Tang [2] ✉

The hexosamine biosynthetic pathway (HBP) produces uridine diphosphate N-acetylglucosamine (UDP-GlcNAc) to facilitate O-linked GlcNAc (O-GlcNAc) protein modifications, and subsequently enhance cell survival under lethal stresses. Transcript induced in spermiogenesis 40 (Tisp40) is an endoplasmic reticulum membrane-resident transcription factor and plays critical roles in cell homeostasis. Here, we show that Tisp40 expression, cleavage and nuclear accumulation are increased by cardiac ischemia/reperfusion (I/R) injury. Global Tisp40 deficiency exacerbates, whereas cardiomyocyte-restricted Tisp40 overexpression ameliorates I/R-induced oxidative stress, apoptosis and acute cardiac injury, and modulates cardiac remodeling and dysfunction following long-term observations in male mice. In addition, overexpression of nuclear Tisp40 is sufficient to attenuate cardiac I/R injury in vivo and in vitro. Mechanistic studies indicate that Tisp40 directly binds to a conserved unfolded protein response element (UPRE) of the glutamine-fructose-6-phosphate transaminase 1 (GFPT1) promoter, and subsequently potentiates HBP flux and O-GlcNAc protein modifications. Moreover, we find that I/R-induced upregulation, cleavage and nuclear accumulation of Tisp40 in the heart are mediated by endoplasmic reticulum stress. Our findings identify Tisp40 as a cardiomyocyte-enriched UPR-associated transcription factor, and targeting Tisp40 may develop effective approaches to mitigate cardiac I/R injury.

Ischemic heart disease (IHD) is one of the leading causes of death worldwide, and imposes a tremendous socioeconomic burden to individuals as well as the healthcare system[1]. Prompt restoration of coronary perfusion is vital to preserve the viable cardiomyocytes around the ischemic foci; however, it also triggers a second wave of cardiac insult, termed ischemia/reperfusion (I/R) injury[2]. To accommodate hypoxic conditions, metabolic machinery and mitochondrial function in the heart are dramatically altered, which, however, drive reactive oxygen species (ROS) overproduction after the sudden influx of nutrients and oxygen during reperfusion, thereby facilitating oxidative damage to the heart[3, 4]. Glucose, an important metabolic substrate for the heart, is primarily catabolized through glycolysis and glycogen synthesis after entering the cells. The hexosamine biosynthetic pathway (HBP) consumes a small but significant fraction of glucose to generate uridine diphosphate N-acetylglucosamine (UDP-GlcNAc), an indispensable nucleotide sugar to govern the highly inducible, dynamic and reversible O-linked GlcNAc (O-GlcNAc) protein modifications[5]. Dysregulated HBP and O-GlcNAcylation have

[1]Department of Geriatrics, Renmin Hospital of Wuhan University, Hubei Key Laboratory of Metabolic and Chronic Diseases, 430060 Wuhan, China. [2]Department of Cardiology, Renmin Hospital of Wuhan University, Hubei Key Laboratory of Metabolic and Chronic Diseases, 430060 Wuhan, China. [3]Hubei Key Laboratory of Metabolic and Chronic Diseases, 430060 Wuhan, China. [4]These authors contributed equally: Xin Zhang, Can Hu. ✉ e-mail: qztang@whu.edu.cn

been implicated in various cardiovascular diseases (e.g., cardiac hypertrophy, heart failure, diabetic cardiomyopathy and cardiac I/R injury), and acute stimulation of HBP and O-GlcNAcylation confers notable cardioprotection against lethal stresses[6–9]. Accordingly, Liu et al. demonstrated that increasing protein O-GlcNAc levels either at pre-ischemic stage or during reperfusion could reduce cardiac injury and dysfunction in the isolated perfused heart[10,11]. Wang et al. revealed that stimulating HBP and O-GlcNAcylation significantly alleviated cardiac I/R injury in mice[12]. Meanwhile, augmentation of O-GlcNAc signaling also prevented I/R-induced mitochondrial damage and ROS accumulation in the heart[13,14]. Therefore, targeting HBP and O-GlcNAcylation may be of great therapeutic interest to treat cardiac I/R injury.

Transcript induced in spermiogenesis 40 (Tisp40, also known as AlbZIP or ATCE1), encoded by *Creb3l4* gene, is an endoplasmic reticulum (ER) membrane-resident type II transmembrane protein, and exhibits two isoforms named Tisp40α and Tisp40β[15–17]. Tisp40α lacks the transcription activation domain (TAD) in its N-terminal portion, and is the inactive form with very low expression. Yet, Tisp40β contains an N-terminal acidic TAD, and acts as the active and predominant form[15]. Therefore, we use the term Tisp40 to mean Tisp40β in this article, unless specifically stated. Under physiological conditions, Tisp40 is anchored to the ER through its transmembrane (TM) domain, and then released to the nucleus by a "regulated intramembrane proteolysis (Rip)" cleavage mechanism, where it functions as an active transcription factor[15,17]. Previous studies suggested that Tisp40 was required for spermatogenesis but not fertility in mice, and that disruption of Tisp40 promoted apoptosis of murine germ cells[18,19]. Tisp40 ablation also accelerated adipocyte differentiation and hyperplasia, thereby improving glucose tolerance and insulin sensitivity in high fat diet (HFD)-treated mice[20]. In addition, Tisp40 played critical roles in the survival of various cancer cells and closely correlated with the prognosis of these patients[21,22]. However, the role and molecular basis of Tisp40 in cardiac I/R injury remains elusive.

Here, we show that Tisp40 expression, cleavage and nuclear accumulation are increased by cardiac I/R injury. Gain- and loss-of-function findings identify Tisp40 as a negative regulator of I/R-induced acute cardiac injury, ventricular remodeling and dysfunction. In addition, overexpression of nuclear Tisp40 is sufficient to attenuate cardiac I/R injury in vivo and in vitro. Mechanistically, ER membrane-resident Tisp40 in I/R-injured hearts is cleaved under ER stress, and then released to the nucleus, where it directly binds to the promoter of GFPT1 and subsequently facilitates HBP flux and protein O-GlcNAcylation, thereby mitigating cardiac I/R injury.

## Results

### Tisp40 expression and nuclear translocation are induced by cardiac I/R injury

Full-length Tisp40 is composed of 370 amino acids with a calculated molecular weight of 41 kDa, and contains an N-terminal acidic TAD in the cytosol, a basic zipper (bZIP) transcription domain and a TM domain, which can be released to the nucleus by a Rip cleavage mechanism at the S1P and S2P sites (Fig. 1a)[15,17]. Previous studies have demonstrated that Tisp40 is highly expressed in murine postmeiotic testis and prostate under physiological conditions, but displays limited abundance in other tissues, including the heart[15,18]. As expected, we detected very low Tisp40 expression in sham-operated hearts; however, its protein level was dramatically elevated in response to I/R injury (Fig. 1b). Intriguingly, a new protein band (lower bands shown by an arrow in lanes 3–4) was observed in whole heart lysates from I/R mice, but not in those from control mice. To clarify whether this new band is the N-terminal fragment of Tisp40, nuclear extracts of heart samples were prepared from sham- or I/R-operated mice. As shown in Fig. 1b, I/R injury dramatically induced the cleavage and nuclear translocation of Tisp40 in the heart. Of note, the specificity of this

Tisp40 antibody was confirmed in Tisp40 knockout (KO) hearts after I/R injury, as it failed to detect full-length or nuclear Tisp40. Meanwhile, double immunofluorescence staining for Tisp40 and sarcomeric α-actinin showed that Tisp40 staining in the heart was very weak at baseline, intensifying after I/R surgery, and that the increased Tisp40 was mainly derived from cardiomyocytes and resided in the nucleus (Fig. 1c). PCR analysis also demonstrated low Tisp40 expression in all cardiac cell populations of sham-operated mice. I/R injury did not affect Tisp40 expression in cardiac fibroblasts, modestly increased Tisp40 expression in macrophages and dramatically elevated Tisp40 expression in cardiomyocytes (Supplementary Fig. 1A). Consistent with in vivo findings, Tisp40 expression and nuclear accumulation were also induced in simulated I/R (sI/R)-treated neonatal rat cardiomyocytes (NRCMs) (Fig. 1d, e). Co-localization of Tisp40 and Lys−Asp−Glu−Leu (KDEL) in NRCMs indicated that Tisp40 mainly localized to the ER at baseline (Fig. 1d). To enhance the clinical impact of our findings, we finally investigated the expression and subcellular distribution of Tisp40 in human hearts and cardiomyocytes. As shown Fig. 1f, Tisp40 expression, cleavage and nuclear translocation were significantly provoked in the left ventricles of IHD patients. In addition, Tisp40 was weakly expressed only in the cytoplasmic extracts of human-induced pluripotent stem cell-derived cardiomyocytes (hiPSC-CMs) at baseline, but upregulated and translocated to the nucleus after sI/R stimulation (Fig. 1g and Supplementary Fig. 1B). Taken together, these results imply that Tisp40 expression and nuclear translocation are induced by cardiac I/R injury.

### Tisp40 deficiency exacerbates oxidative stress, apoptosis, and cardiac I/R injury in vivo and in vitro

To investigate the role of Tisp40 in cardiac I/R injury, global Tisp40 KO mice were established, and Tisp40 deficiency was validated by western blot as well as genotyping methods (Fig. 1b and Supplementary Fig. 2A). As shown in Supplementary Fig. 2B, Tisp40 ablation did not alter *Creb3l1*, *Creb3l2*, or *Creb3l3* mRNA levels in the heart. Consistent with previous observations, Tisp40 KO mice were indistinguishable from their wild-type (WT) littermates in appearance, gross behavior and long-term survival rates. Kim et al. revealed that Tisp40 played critical roles in maintaining glucose homeostasis of HFD-treated mice; however, no overt metabolic disorders were observed in non-stressed Tisp40 KO mice in our study (Supplementary Table 3)[20]. Meanwhile, Tisp40-deficient mice manifested no pathological alterations in cardiac structure or function under basal conditions, at least within 12 months (Supplementary Fig. 2C). Yet, they exhibited larger infarct area (IA) upon cardiac I/R injury, despite a similar surgical injury as the WT littermates (Fig. 2a–c). Serum levels of cardiac isoform of tropnin T (cTnT), creatine kinase isoenzymes (CK-MB) and lactate dehydrogenase (LDH) were also increased in Tisp40 KO mice 4 h after cardiac I/R surgery (Supplementary Fig. 3A). Meanwhile, Tisp40 deficiency dramatically facilitated cell apoptosis after I/R surgery, as evidenced by the increased TdT-mediated dUTP nick end-labeling (TUNEL)-positive nuclei and formation of DNA fragments (Fig. 2d, e). Consistently, B-cell lymphoma-2 (BCL-2) protein was downregulated, while BCL-2-associated X protein (BAX) expression was upregulated in Tisp40 KO hearts upon I/R injury (Supplementary Fig. 3B). And caspase3 activity in I/R-injured hearts was further elevated by Tisp40 deletion (Fig. 2f). ROS overproduction is a key feature of cardiac I/R injury, and contributes to cell apoptosis and cardiac dysfunction. As shown in Fig. 2g, h and Supplementary Fig. 3C, I/R-induced elevations of superoxide anion ($O_2^-$) and hydrogen peroxide ($H_2O_2$) were further enhanced in Tisp40 KO hearts, accompanied by increased generations of 3-nitrotyrosine (3-NT), malondialdehyde (MDA) and 4-hydroxynonenal (4-HNE). Of note, no significant difference in mortality rate was observed between the two groups during 24 h of follow-up after cardiac I/R injury (8.0% in WT mice vs. 12.0% in KO mice, $P > 0.05$).

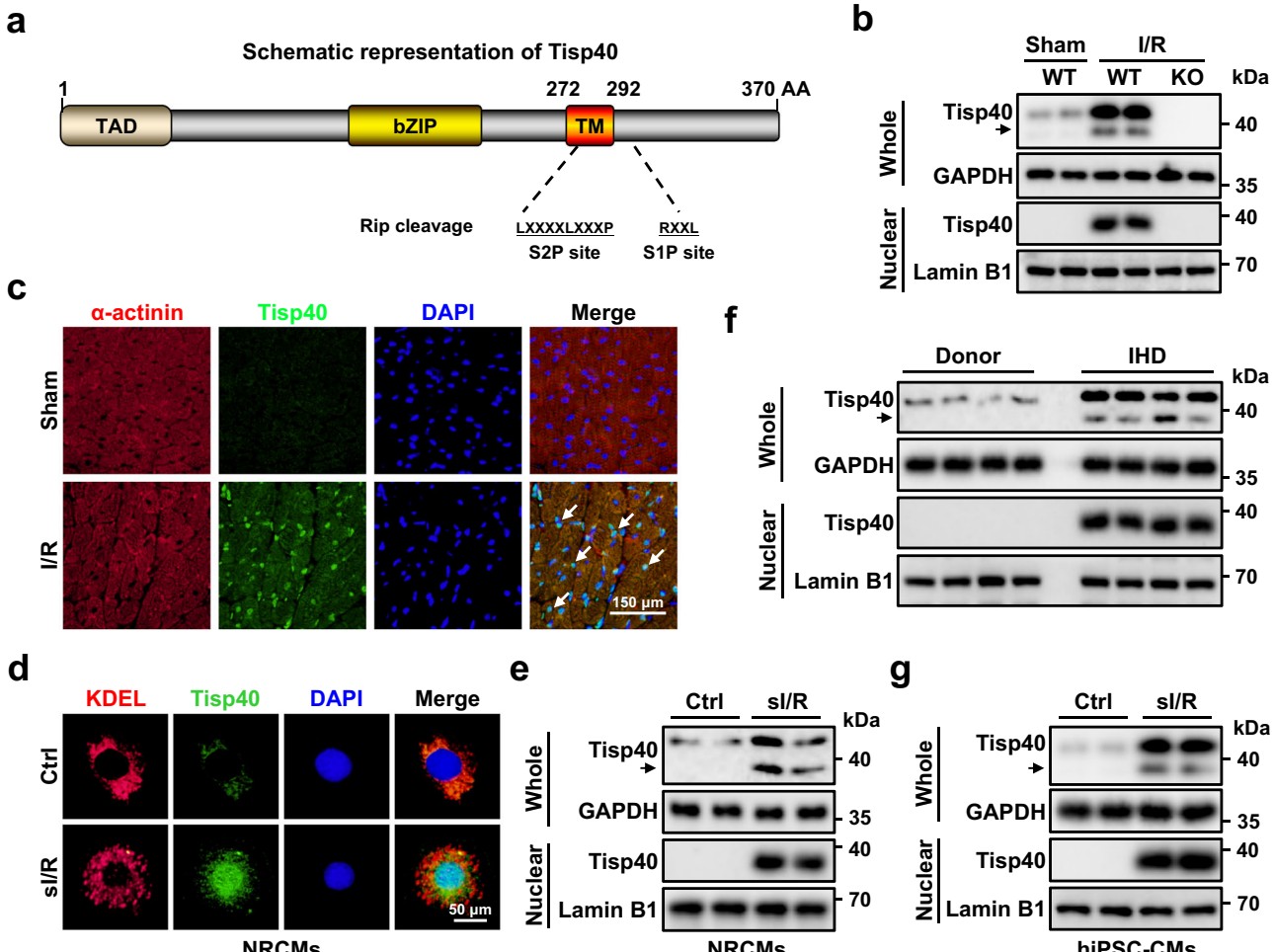

**Fig. 1 | Tisp40 expression and nuclear translocation are induced by cardiac I/R injury. a** Full-length Tisp40 (370 amino acids) is an endoplasmic reticulum (ER) membrane-resident type II transmembrane protein, and contains an N-terminal acidic transcription activation domain (TAD), a basic zipper (bZIP) domain and a transmembrane (TM) domain, which is cleaved by S1P and S2P proteases to release the N-terminal fragment to the nucleus. **b** Tisp40 knockout (KO) mice and the wild-type (WT) littermates received sham or cardiac ischemia/reperfusion (I/R) surgery (ischemia for 45 min and reperfusion for 24 h), and then whole-cell lysates together with nuclear lysates from the heart were prepared for western blot ($n = 6$). **c** Heart samples were collected for immunofluorescence staining of sarcomeric α-actinin (red) and Tisp40 (green) ($n = 6$). **d** usion, and then were stained with Lys–Asp–Glu–Leu (KDEL, an ER marker, red) and Tisp40 (green) ($n = 6$). **e** Whole-cell lysates and nuclear lysates from NRCMs were prepared for western blot ($n = 6$). **f** Whole-cell lysates and nuclear lysates from the left ventricles of ischemic heart disease (IHD) patients or donors were prepared for western blot ($n = 6$). **g** Human-induced pluripotent stem cell-derived cardiomyocytes (hiPSC-CMs) were exposed to ischemia for 4 h followed by overnight reperfusion, and then whole-cell lysates together with nuclear lysates were prepared for western blot ($n = 6$).

To verify the role of Tisp40 in vitro, NRCMs were infected with two independent lentivirus-delivered shRNAs to stably knock down endogenous Tisp40. Western blot showed that both full-length and cleaved Tisp40 in sI/R-treated NRCMs were downregulated by shTisp40 or shTisp40# (Supplementary Fig. 4A). As shown in Supplementary Fig. 4B–D, Tisp40 silence significantly increased TUNEL-positive nuclei and formation of DNA fragments in sI/R-stimulated NRCMs. Accordingly, BAX protein and caspase3 activity were increased, while BCL-2 protein was decreased in shTisp40-infected NRCMs after sI/R (Supplementary Fig. 4E, F). Meanwhile, we found that sI/R-induced oxidative stress was also exacerbated in Tisp40-deficient NRCMs (Supplementary Fig. 4G, H). As expected, Tisp40 knockdown further decreased cell viability and increased LDH releases in sI/R-stimulated NRCMs (Supplementary Fig. 4I, J). In addition, we also used an independent shTisp40# to confirm specificity and exclude any off-target effects. Consistently, sI/R-induced oxidative stress and cardiomyocyte apoptosis were also exacerbated in shTisp40#-infected NRCMs (Supplementary Fig. 5A–F). Collectively, we prove that Tisp40 deficiency exacerbates oxidative stress, apoptosis and cardiac I/R injury in vivo and in vitro.

## Tisp40 deficiency aggravates cardiac remodeling and dysfunction following I/R injury

The above findings revealed that Tisp40 ablation exacerbated I/R-induced acute cardiac injury, and we then investigated whether Tisp40 deficiency elicited long-term structural or functional alterations to I/R-stressed hearts. Compared with WT littermates, Tisp40 KO mice displayed a significantly greater increase in cardiomyocyte size and heart weight/tibial length (HW/TL) 4 weeks post-I/R injury, accompanied by a re-activation of cardiac fetal gene programs, as determined by the increased atrial natriuretic peptide (*Anp*), *β-Mhc* and decreased *α-Mhc* mRNA levels (Fig. 3a, b and Supplementary Fig. 6A). And the fibrotic area was also increased in Tisp40-deficient hearts (Fig. 3c and Supplementary Fig. 6B). In addition, Tisp40 deficiency dramatically increased the expression of α-smooth muscle actin (α-SMA), a landmark for the fibroblast-to-myofibroblast transition, in I/R-stressed hearts (Supplementary Fig. 6C). Consistently, mRNA levels of fibrotic markers, that is, collagen 1α1 (*Col1α1*), *Col3α1*, transforming growth factor-β1 (*Tgf-β1*) and connective tissue growth factor (*Ctgf*), in I/R-injured hearts were further elevated by Tisp40 ablation (Supplementary Fig. 6D). Cardiac fibrosis is characterized, not only by excessive accumulation of

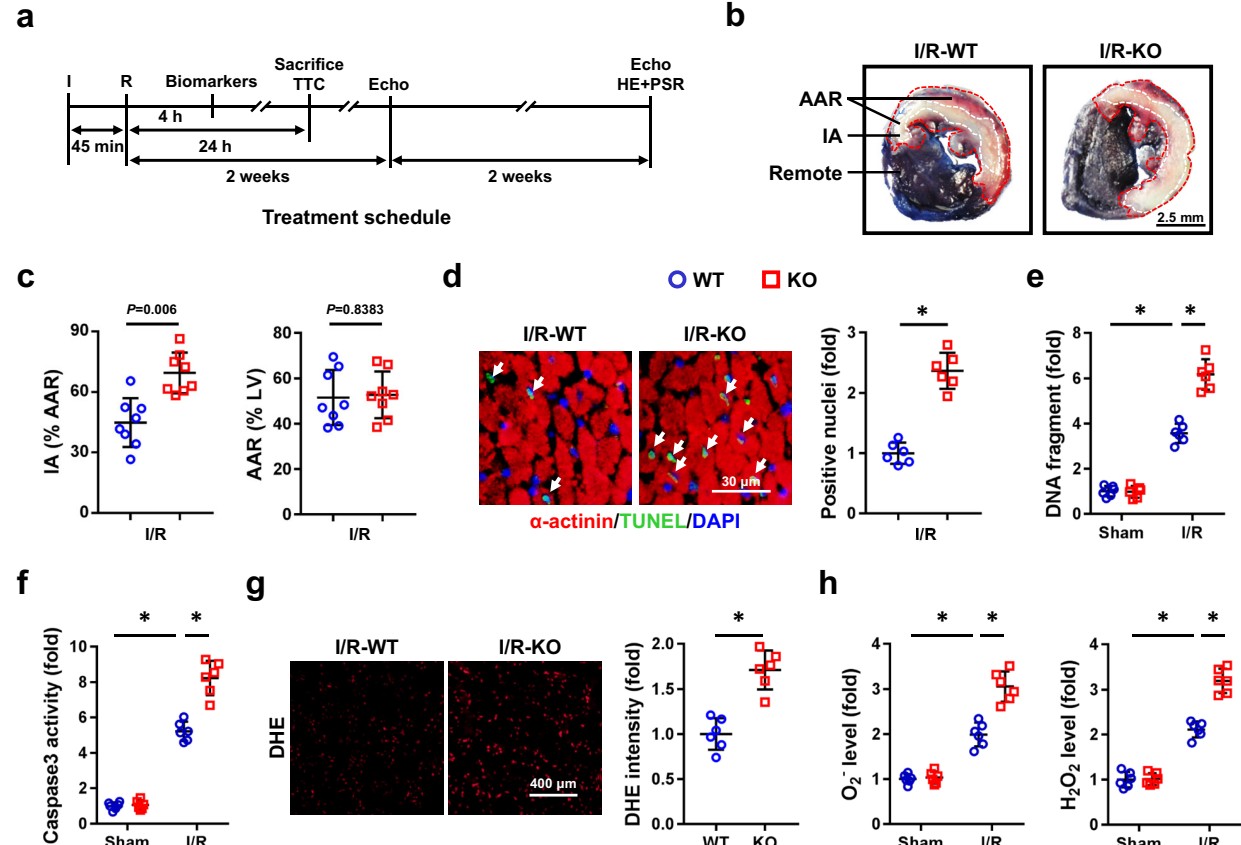

**Fig. 2 | Tisp40 deficiency exacerbates oxidative stress, apoptosis, and cardiac I/R injury in vivo. a** Protocol diagram for the study design. **b** Representative Evans blue and 2,3,5-triphenyltetrazolium chloride (TTC)-stained heart sections from global Tisp40 KO mice and WT littermates (*n* = 8). **c** The relative ratios of infarct area (IA, pale) to the area at risk (AAR, not blue) and AAR to left ventricles (LV) were compared between Tisp40 KO and WT hearts 24 h after I/R surgery (*n* = 8). **d** Representative TdT-mediated dUTP nick end-labeling (TUNEL) staining images of heart sections and quantitative results from Tisp40 KO mice and WT littermates

24 h after I/R surgery (*n* = 6). **e** DNA fragments in the heart (*n* = 6). **f** Caspase3 activity in the heart (*n* = 6). **g** Representative dihydroethidium (DHE) staining images of heart sections and quantitative results from Tisp40 KO mice and WT littermates 24 h after I/R surgery (*n* = 6). **h** Quantitative results of superoxide anion ($O_2^-$) and hydrogen peroxide ($H_2O_2$) in the heart (*n* = 6). All data are expressed as the mean ± SD, and analyzed using an unpaired two-tailed Student's *t* test. For the analysis in (**e, f, h**), one-way analysis of variance (ANOVA) followed by Tukey post hoc test was conducted. \**P* < 0.0001. Source data are provided as a Source Data file.

extracellular matrix proteins, but also by increased cross-linking of collagen fibrils within the fibers. Lysyl oxidase (LOX), a copper-dependent amine oxidase, catalyzes the cross-linking of extracellular matrix proteins to form insoluble fibers that possess increased thickness and material stiffness[23]. As shown in Fig. 3d and Supplementary Fig. 6E, F, the levels of *Lox* mRNA and insoluble collagen were significantly increased in I/R-injured hearts but to a more prominent extent in those with Tisp40 deletion; however, no alteration in soluble collagen content was observed. To test for functional relevance, we measured cardiac function at baseline, 2 and 4 weeks post-I/R injury. At baseline, the contractile function and morphology of Tisp40 KO hearts did not differ from the WT hearts. Yet, Tisp40 deficiency aggravated I/R-induced cardiac dysfunction and ventricular dilation, as evidenced by the decreased fractional shortening (FS) and increased left ventricle internal diameters at diastole (LVIDd) and left ventricle internal diameters at systole (LVIDs) post-I/R surgery (Fig. 3e, f). In addition, echocardiographic analysis also revealed profound cardiac hypertrophy in I/R-injured Tisp40 KO mice, as determined by the increased interventricular septal thickness at systole (IVSs) but not interventricular septal thickness at diastole (IVSd) (Fig. 3g and Supplementary Fig. 6G). Of note, no difference in heart rate (HR) was identified between groups before- or post-I/R surgery (Supplementary Fig. 6H).

As Tisp40 was modestly upregulated in cardiac macrophages after I/R surgery, we then performed bone marrow transplantation

study to investigate whether macrophage Tisp40 provided cardioprotection against I/R injury. Successful generation of bone marrow chimera was verified by PCR analysis of genomic DNA from peripheral blood (Supplementary Fig. 7A). As shown in Supplementary Fig. 7B, C, Tisp40 KO-recipient mice displayed similar IA and acute cardiac injury post-I/R surgery regardless of bone marrow origin. In addition, reconstituting either WT or KO bone marrows in Tisp40 KO mice did not affect cardiomyocyte hypertrophy and fibrotic remodeling 4 weeks post-I/R surgery (Supplementary Fig. 7D, E). Accordingly, no significant differences in FS, LVIDd and LVIDs were found in Tisp40 KO mice receiving either WT or KO bone marrows (Supplementary Fig. 7F, G). Collectively, these loss-of-function findings suggest that cardiomyocyte Tisp40 is required for preventing I/R-induced acute cardiac injury, ventricular remodeling and dysfunction in mice.

**Cardiomyocyte-specific overexpression of full-length Tisp40 prevents I/R-induced acute cardiac injury, remodeling, and dysfunction in mice**

In view of the fact that cardiomyocytes were the primary cell populations expressing Tisp40 upon I/R injury and that macrophage Tisp40 did not affect cardiac I/R injury, we then generated cardiomyocyte-restricted Tisp40 transgenic (cTG) mice to determine whether overexpressing Tisp40 in cardiomyocytes could prevent cardiac I/R injury. Four germ lines of Tisp40 cTG mice were established, and we selected

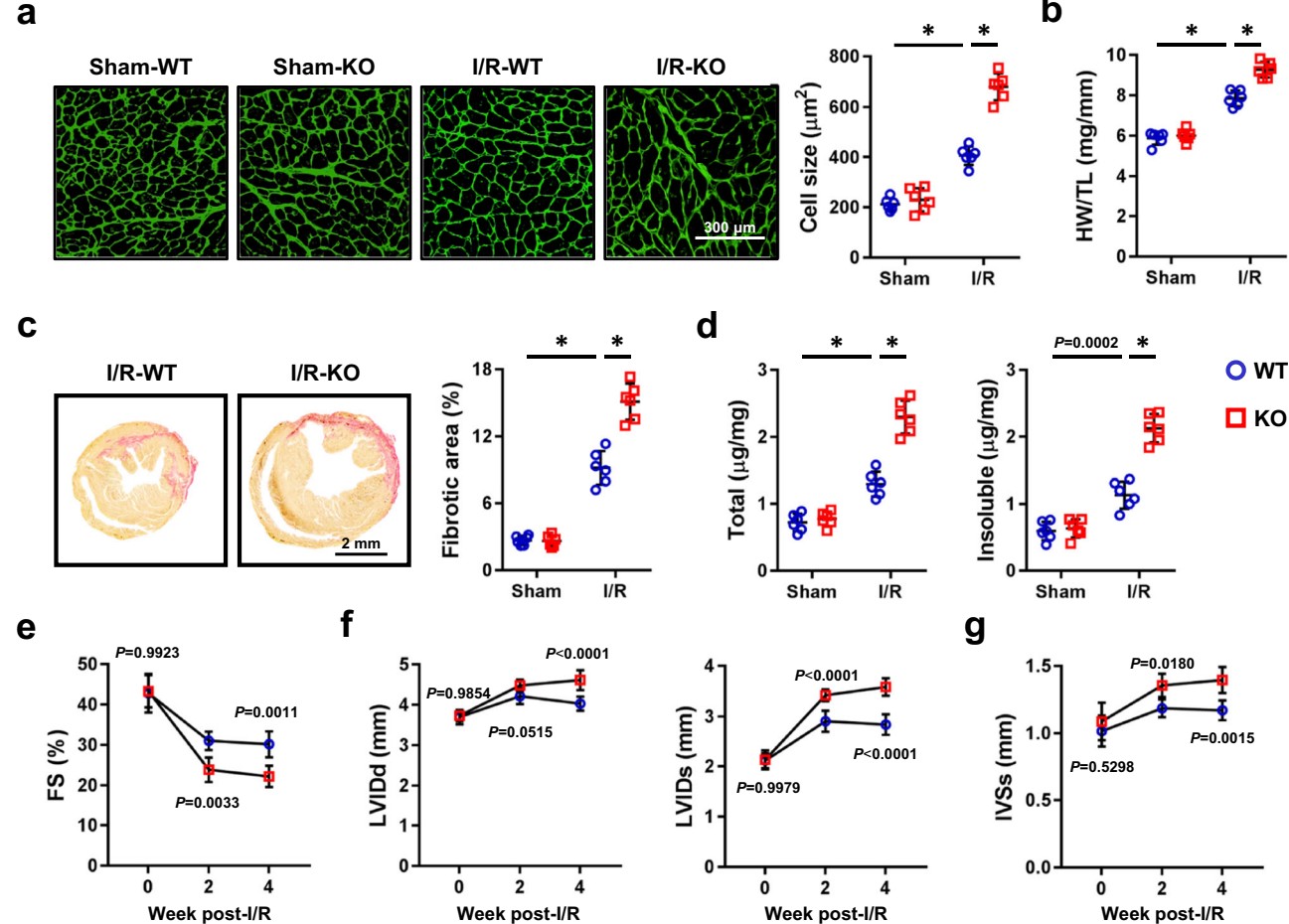

**Fig. 3 | Tisp40 deficiency aggravates cardiac remodeling and dysfunction following I/R injury. a** Heart samples were collected for Wheat Germ Agglutinin (WGA) staining to quantify the cross-sectional area of cardiomyocyte 4 weeks post-I/R surgery (*n* = 6). **b** Quantitative results of heart weight/tibial length (HW/TL) 4 weeks post-I/R surgery (*n* = 6). **c** Heart samples were collected for picrosirius red (PSR) staining to quantify the collagen deposition 4 weeks post-I/R surgery (*n* = 6). **d** Total and insoluble collagen content in the heart 4 weeks post-I/R surgery (*n* = 6). **e**–**g** Cardiac function of Tisp40 KO mice or WT littermates was analyzed by transthoracic echocardiography at the indicated time points, and presented as fractional shortening (FS), left ventricle internal diameters at diastole (LVIDd), systole (LVIDs) or interventricular septal thickness at systole (IVSs) (*n* = 6). All data are expressed as the mean ± SD, and analyzed using one-way ANOVA followed by Tukey post hoc test. For the analysis in (**e**–**g**), repeated measures ANOVA followed by Sidak post hoc test was conducted. *\*P* < 0.0001. Source data are provided as a Source Data file.

line 4, which expressed the highest levels of Tisp40, for further investigation. Intriguingly, overexpressing full-length Tisp40 in the heart was accompanied by the cleavage and nuclear accumulation of Tisp40 even without I/R stress, indicating the existence of a background cleavage when Tisp40 is strongly overexpressed (Supplementary Fig. 8A). Tisp40 overexpression did not affect fasting blood glucose, serum insulin, homeostatic model assessment–insulin resistance (HOMA-IR) index, serum triglyceride and total cholesterol levels in mice (Supplementary Table 3). Yet, it significantly ameliorated I/R-induced acute cardiac injury, as demonstrated by the smaller IA and decreased serum levels of cTnT, CK-MB, and LDH (Fig. 4a and Supplementary Fig. 8B). Cell apoptosis in I/R-stressed hearts was also inhibited by Tisp40 overexpression (Fig. 4b–d and Supplementary Fig. 8C). Meanwhile, overexpression of Tisp40 in cardiomyocytes significantly reduced the levels of cardiac $O_2^-$ and $H_2O_2$ in I/R-stressed hearts and suppressed ROS toxicity to lipids and proteins (Fig. 4e, f and Supplementary Fig. 8D). Of note, neither ablation nor overexpression of Tisp40 affected ROS generation and cell apoptosis in non-stressed hearts. Consistent with the improved acute cardiac injury, Tisp40 cTG mice developed significantly attenuated cardiomyocyte hypertrophic growth 4 weeks post-I/R injury (Fig. 4g and Supplementary Fig. 8E, F). In addition, Tisp40 overexpression also inhibited I/R-induced

myofibroblast activation and fibrotic remodeling, as evidenced by the decreased fibrotic area, α-SMA expression, and *Col1α1*, *Col3α1*, *Tgf-β1*, *Ctgf* mRNA levels (Fig. 4h and Supplementary Fig. 8G–I). Moreover, both *Lox* mRNA level and collagen cross-linking in I/R-stressed hearts were suppressed by Tisp40 overexpression (Fig. 4i, j and Supplementary Fig. 8J). Accordingly, Tisp40 cTG mice also developed an improved cardiac function following I/R surgery (Fig. 4k and Supplementary Fig. 8K). In aggregate, these gain-of-function results demonstrate that cardiomyocyte-specific overexpression of full-length Tisp40 can prevent I/R-induced acute cardiac injury, remodeling and dysfunction in mice.

## Overexpression of nuclear Tisp40 is sufficient to attenuate cardiac I/R injury in vivo and in vitro

The aforementioned results revealed that Tisp40 cleavage and nuclear accumulation were induced in both I/R-injured control hearts and unstressed Tisp40 cTG hearts; therefore, we next investigated whether this cleaved Tisp40 in the nucleus was the active form to prevent cardiac I/R injury. To address this hypothesis, Tisp40 KO mice were exposed to a single intravenous injection of AAV9ΔTM-HA 4 weeks prior to I/R surgery, and abundant Tisp40 expression was noted in the nucleus of cardiomyocytes (Fig. 5a, b). As shown in Fig. 5c, d,

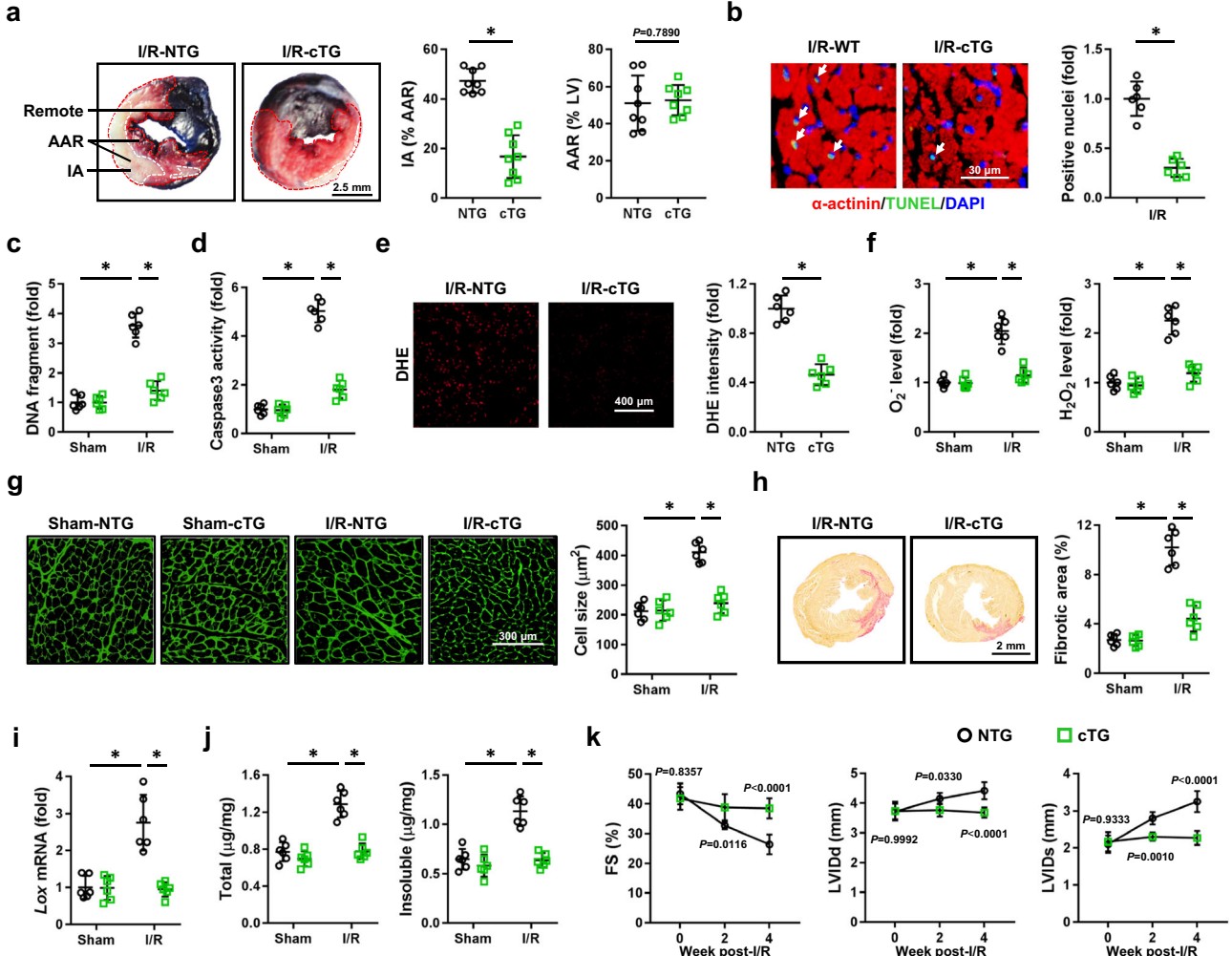

**Fig. 4 | Cardiomyocyte-specific overexpression of full-length Tisp40 prevents I/R-induced acute cardiac injury, remodeling, and dysfunction in mice.**
**a** Representative Evans blue and TTC-stained heart sections, and quantitative data from cardiomyocyte-restricted Tisp40 transgenic (cTG) mice and the matched non-transgenic (NTG) littermates 24 h after I/R surgery (*n* = 8). **b** Representative TUNEL staining images of heart sections and quantitative results from Tisp40 cTG mice and WT littermates 24 h after I/R surgery (*n* = 6). **c** DNA fragments in the heart (*n* = 6). **d** Caspase3 activity in the heart (*n* = 6). **e** Representative DHE staining images of heart sections and quantitative results from Tisp40 cTG mice and WT littermates 24 h after I/R surgery (*n* = 6). **f** Quantitative results of $O_2^-$ and $H_2O_2$ in the heart (*n* = 6). **g** Heart samples were collected for WGA staining to quantify the cross-

sectional area of cardiomyocytes 4 weeks post-I/R surgery (*n* = 6). **h** Heart samples were collected for PSR staining to quantify the collagen deposition 4 weeks post-I/R surgery (*n* = 6). **i** Levels of lysyl oxidase (*Lox*) mRNA in the heart 4 weeks post-I/R surgery (*n* = 6). **j** Total and insoluble collagen content in the heart 4 weeks post-I/R surgery (*n* = 6). **k** Cardiac function of Tisp40 cTG mice or NTG littermates was analyzed by transthoracic echocardiography at the indicated time points, and presented as FS, LVIDd and LVIDs (*n* = 6). All data are expressed as the mean ± SD, and analyzed using one-way ANOVA followed by Tukey post hoc test. For the analysis in (**a**, **b**, **e**), an unpaired two-tailed Student's *t* test was conducted. For the analysis in (**k**), repeated measures ANOVA followed by Sidak post hoc test was conducted. *\*P* < 0.0001. Source data are provided as a Source Data file.

I/R-induced increases of IA and serum cTnT, CK-MB and LDH levels in Tisp40 KO hearts were significantly reduced after the overexpression of nuclear Tisp40. In addition, overexpression of nuclear Tisp40 also ameliorated cardiac hypertrophy and fibrosis seen in Tisp40 KO hearts upon I/R injury (Fig. 5e–i). Consistently, Tisp40 KO mice expressing nuclear Tisp40 exhibited better cardiac function upon I/R insult, as evidenced by the increased FS and decreased LVIDd and LVIDs (Fig. 5j, k). To further determine the role of nuclear Tisp40 specifically in cardiomyocytes, NRCMs were infected with AdΔTM-HA, and the efficiency was validated by immunofluorescence staining and western blot (Supplementary Fig. 9A, B). As shown in Supplementary Fig. 9C, D, overexpression of nuclear Tisp40 significantly attenuated sI/R-induced cardiomyocyte injury and death, as indicated by the increased cell viability and decreased LDH releases. In addition, AdΔTM-HA infection also reduced caspase3 activity and BAX level, but restored BCL-2 expression in sI/R-treated NRCMs (Supplementary Fig. 9E, F). Accordingly, sI/R-induced apoptosis of NRCMs was

effectively inhibited in those overexpressed with nuclear Tisp40, as evidenced by the decreased TUNEL-positive nuclei and formation of DNA fragments (Supplementary Fig. 9G, H). Meanwhile, ROS production in sI/R-treated NRCMs was also suppressed by AdΔTM-HA infection, accompanied by the reductions of MDA, 4-HNE, and 3-NT (Supplementary Fig. 9I, J). Collectively, these findings indicate that Tisp40 upregulation and nuclear translocation in cardiomyocytes are compensatory self-protective responses, and that overexpression of nuclear Tisp40 is sufficient to attenuate cardiac I/R injury in vivo and in vitro.

## Tisp40 ameliorates cardiac I/R injury through stimulating HBP flux and protein O-GlcNAcylation

To determine the molecular basis through which Tisp40 ameliorates cardiac I/R injury, we re-analyzed the microarray data from human prostate cells expressing a conditionally active Tisp40[24]. Kyoto Encyclopedia of Genes and Genomes (KEGG) analysis revealed that the HBP

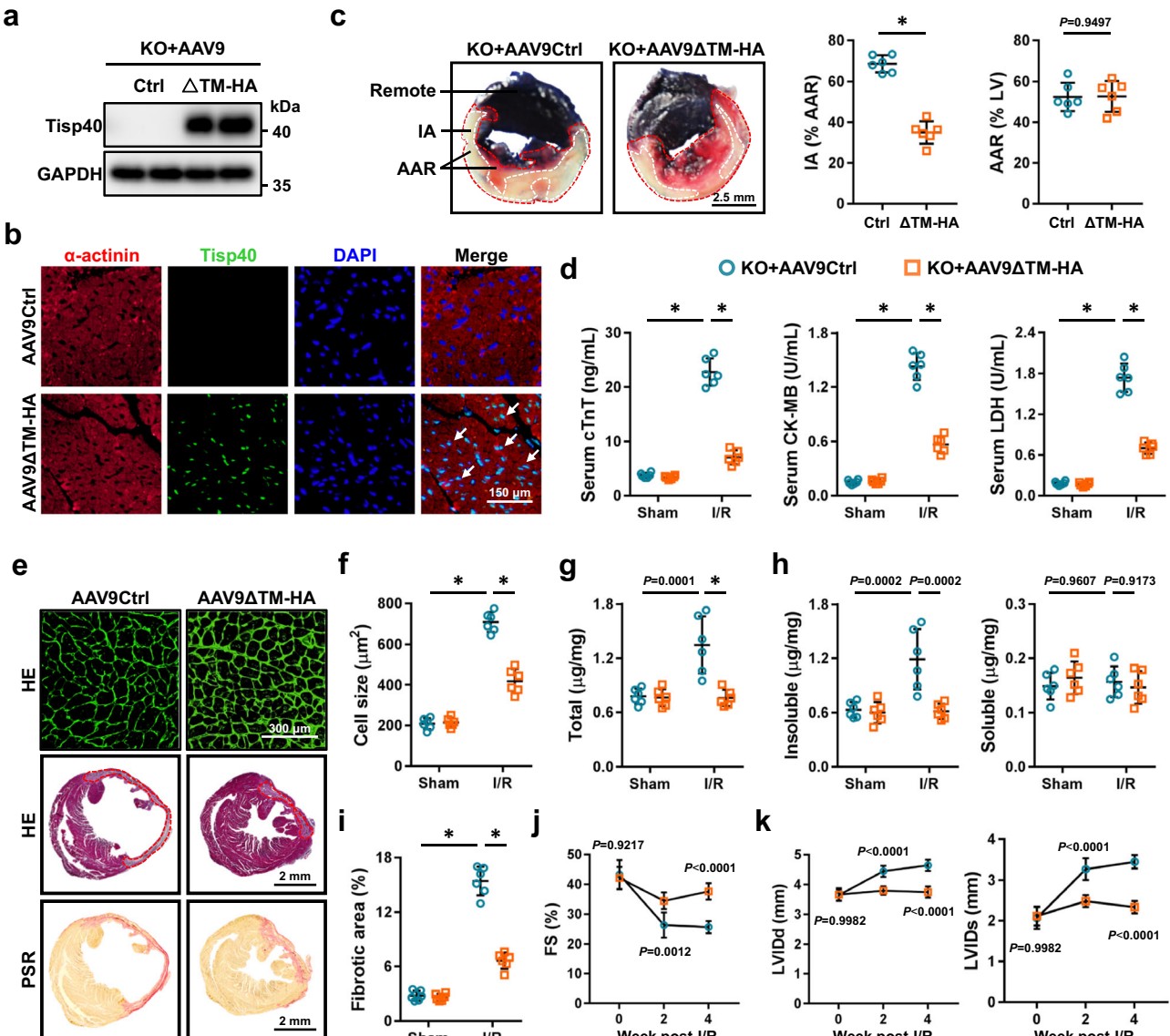

**Fig. 5 | Overexpression of nuclear Tisp40 is sufficient to attenuate cardiac I/R injury in vivo. a** Tisp40 KO mice received a single intravenous injection of AAV9ΔTM-HA or AAV9Ctrl, and heart samples were collected for western blot 4 weeks post-AAV9 injection (n = 6). **b** Heart samples were collected for immunofluorescence staining of sarcomeric α-actinin (red) and Tisp40 (green) (n = 6). **c** Representative Evans blue and TTC-stained heart sections, and quantitative data from Tisp40 KO mice with AAV9ΔTM-HA or AAV9Ctrl injection 24 h after I/R surgery (n = 6). **d** Circulating levels of cTnT, CK-MB and LDH in Tisp40 KO mice with AAV9ΔTM-HA or AAV9Ctrl injection 4 h after I/R surgery (n = 6). **e** Heart samples

were collected for HE or PSR staining 4 weeks post-I/R surgery (n = 6). **f** Quantitative results of the cross-sectional area of cardiomyocyte (n = 6). **g, h** Total, insoluble and soluble collagen content in the heart 4 weeks post-I/R surgery (n = 6). **i** Quantitative results of the collagen deposition (n = 6). **j, k** Cardiac function was presented as FS, LVIDd, and LVIDs (n = 6). All data are expressed as the mean ± SD, and analyzed using one-way ANOVA followed by Tukey post hoc test. For the analysis in (**c**), an unpaired two-tailed Student's t test was conducted. For the analysis in (**j, k**), repeated measures ANOVA followed by the Sidak post hoc test was conducted. *P < 0.0001. Source data are provided as a Source Data file.

was dramatically activated by Tisp40, accompanied by an increased N-glycan biosynthesis (Fig. 6a). The HBP generates UDP-GlcNAc not only for glycan synthesis, but also for protein O-GlcNAcylation (Fig. 6b). Interestingly, we found that O-GlcNAc protein modification was increased in Tisp40 cTG hearts and AdΔTM-HA-infected NRCMs at baseline (Supplementary Fig. 10A, B). In contrast, Tisp40 deficiency effectively inhibited HBP flux and protein O-GlcNAcylation in I/R-stressed hearts (Fig. 6c–g). To validate whether Tisp40 ameliorated cardiomyocyte injury through stimulating protein O-GlcNAcylation in vitro, sI/R-stimulated NRCMs with or without AdΔTM-HA infection were silenced with O-GlcNAc transferase (OGT) to block protein O-GlcNAcylation. As shown in Supplementary Fig. 10C, D, blocking protein O-GlcNAcylation significantly abolished Tisp40 overexpression-mediated cellular protection against sI/R injury, as indicated by the

decreased cell viability and increased LDH releases. Meanwhile, the inhibitory effects of Tisp40 on sI/R-induced cardiomyocyte apoptosis and oxidative stress were also blunted by OGT silence (Supplementary Fig. 10E–I). Similarly, the increased cell viability and decreased LDH releases seen in AdΔTM-HA-infected NRCMs after sI/R injury were also counteracted by alloxan monohydrate (ALX), an OGT inhibitor (Supplementary Fig. 10J, K). In contrast, the deleterious effects of Tisp40 silence on sI/R-treated NRCMs were prevented either by supplementation of glucosamine hydrochloride (GlcN) and N-acetyl-D-glucosamine (GlcNAc), or by O-GlcNAcase (OGA) inhibition (Supplementary Fig. 11A, B). OGA knockdown also offset Tisp40 silence-mediated aggravation of cellular damage in sI/R-stimulated NRCMs, as evidenced by the increased cell viability and decreased LDH releases (Supplementary Fig. 11C, D). Consistent with in vitro findings, OGA

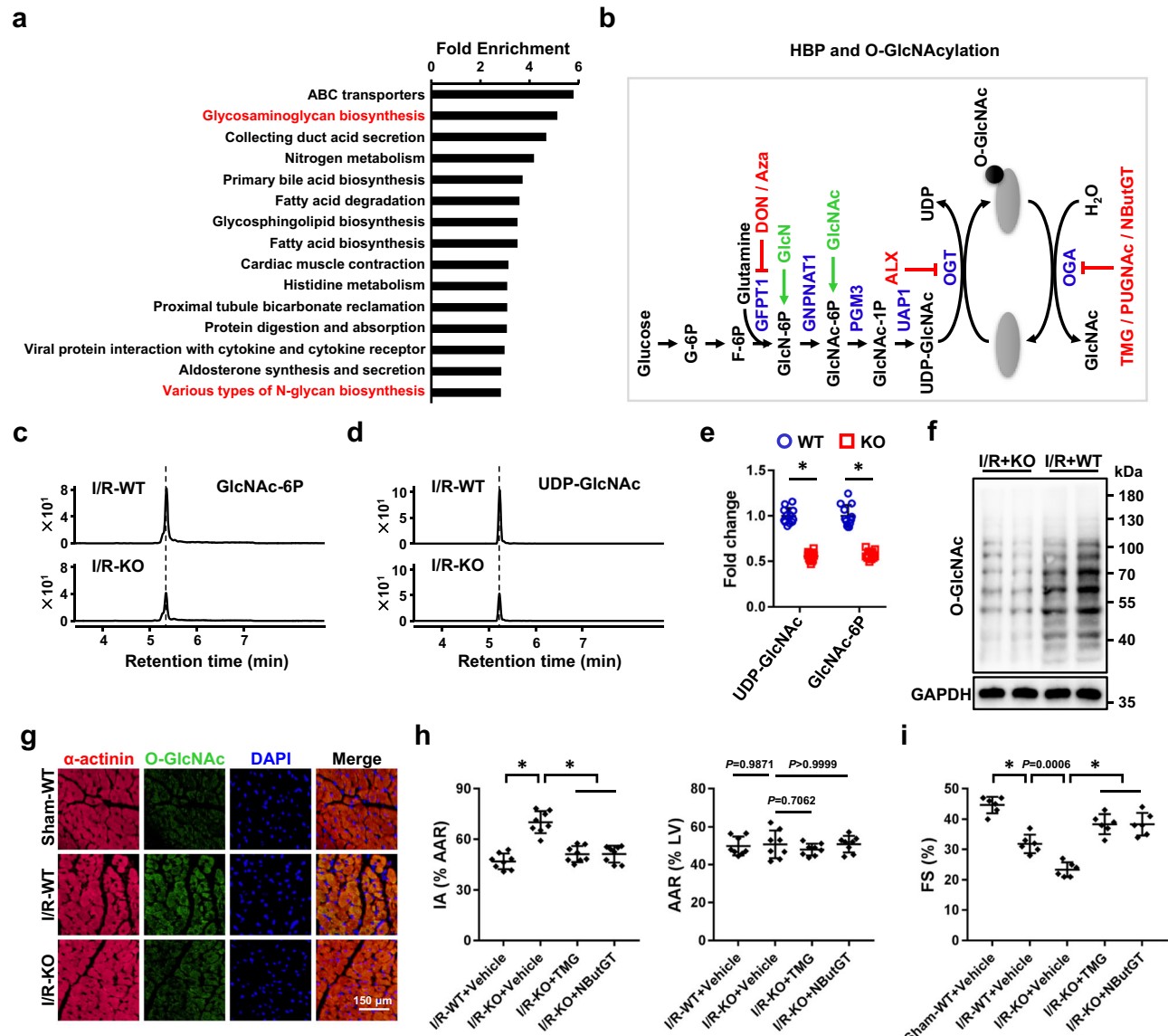

**Fig. 6 | Tisp40 ameliorates cardiac I/R injury through stimulating HBP flux and protein O-GlcNAcylation. a** KEGG analysis of GSE7223 dataset. **b** An overview of the hexosamine biosynthetic pathway (HBP). **c–e** Analysis of GlcNAc-6P and UDP-GlcNAc in I/R-injured Tisp40 KO or WT hearts by UHPLC-MS/MS (n = 12). **f, g** Protein O-GlcNAc levels in I/R-injured Tisp40 KO or WT hearts were evaluated by western blot or by immunofluorescence staining of sarcomeric α-actinin (red) and O-GlcNAc (green) (n = 6). **h** To inhibit OGA, Tisp40 KO mice were intraperitoneally

injected with NButGT daily for 14 consecutive days or TMG every other day for 20 consecutive days, and the last injections of NButGT or TMG were done 30 min before cardiac I/R surgery. Next, Evans blue and TTC staining were performed to demarcate IA and AAR (n = 8). **i** Quantitative results of FS in mice (n = 6). All data are expressed as the mean ± SD, and analyzed using one-way ANOVA followed by Tukey post hoc test. For the analysis in (**e**), an unpaired two-tailed Student's t test was conducted. *P < 0.0001. Source data are provided as a Source Data file.

inhibition with either thiamet G (TMG) or NButGT in vivo also reduced acute cardiac I/R injury in Tisp40 KO hearts (Fig. 6h and Supplementary Fig. 12A). Meanwhile, the increased cell apoptosis and oxidative stress seen in Tisp40 KO hearts post-I/R surgery were significantly suppressed by OGA inhibition (Supplementary Fig. 12B–D). In addition, Tisp40 deficiency-mediated deterioration of cardiomyocyte hypertrophy and fibrotic remodeling post-I/R surgery was also blocked by TMG or NButGT treatment (Supplementary Fig. 12E, F). Accordingly, augmenting protein O-GlcNAcylation by OGA inhibition partially restored the worse cardiac function in Tisp40 KO mice after I/R injury (Fig. 6i and Supplementary Fig. 12G). Previous studies indicated that Tisp40 could bind to a nuclear factor-κB (NF-κB)-binding element, and regulate renal I/R injury[25, 26]. Thus, we also evaluated NF-κB activation and inflammation in Tisp40 KO hearts after I/R. As shown in Supplementary Fig. 13A, Tisp40 deficiency did not affect NF-κB phosphorylation and nuclear

translocation either at baseline or under I/R stress. Accordingly, levels of interleukin-6 (IL-6) and tumor necrosis factor-α (TNF-α) in the heart were also unaffected by Tisp40 ablation (Supplementary Fig. 13B). Collectively, these in vivo and in vitro findings imply that the activation of HBP flux and protein O-GlcNAcylation is involved in Tisp40-mediated cardioprotection against I/R injury.

### Tisp40 facilitates HBP flux and cardioprotection through transcriptionally upregulating GFPT1

We then investigated how Tisp40 regulated HBP flux. Transcription of GFPT1, a rate-limiting enzyme of the HBP, was dramatically increased in human prostate cells expressing a conditionally active Tisp40 (Fig. 7a and Supplementary Fig. 14A)[9,27]. Consistently, we also found that GFPT1, instead of GFPT2, was upregulated in Tisp40 cTG hearts at baseline (Supplementary Fig. 14B, C). Moreover, expression of GFPT1 protein

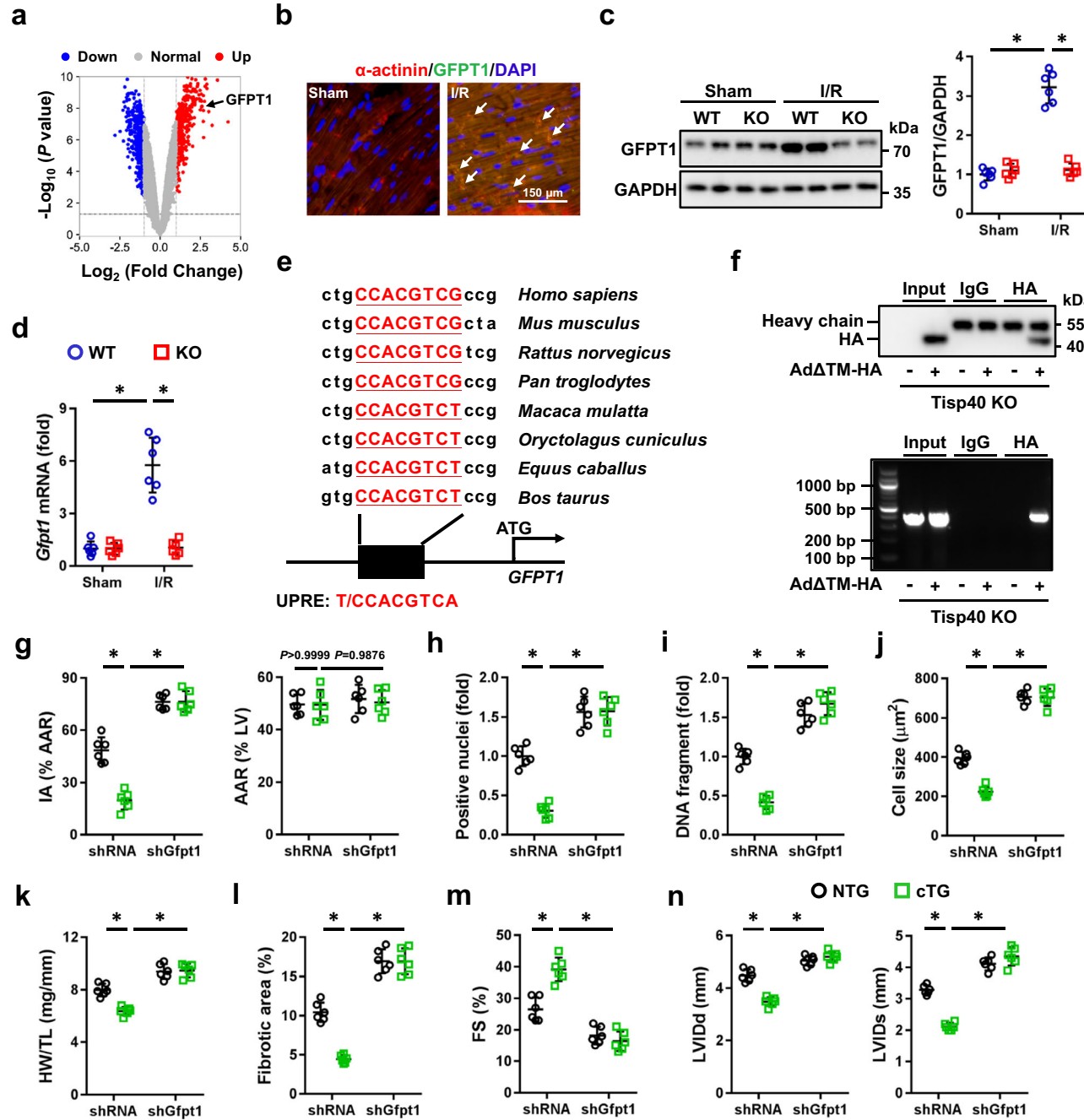

**Fig. 7 | Tisp40 facilitates HBP flux and cardioprotection through transcriptionally upregulating GFPT1. a** Volcano map of GSE7223 dataset. **b** Sham- or I/R-operated hearts were collected for immunofluorescence staining of sarcomeric α-actinin (red) and GFPT1 (green) 24 h after I/R surgery ($n = 6$). **c, d** Heart samples were collected for western blot and quantitative real-time PCR 24 h after I/R surgery ($n = 6$). **e** A conserved UPRE was identified in the GFPT1 promoter across different species. **f** Tisp40-deficient neonatal mouse cardiomyocytes were infected with AdΔTM-HA for 4 h and cultured in fresh medium for an additional 48 h, which were then cross-linked with 1% formaldehyde and immunoprecipitated with anti-HA or anti-IgG isotype control. PCR amplification was performed using primers spanning the UPRE in the GFPT1 promoter ($n = 6$). **g** To knock down endogenous GFPT1 in the heart, Tisp40 cTG mice were intravenously injected with shGfpt1 4 weeks before I/R surgery. Next, Evans blue and TTC staining were performed to demarcate IA and AAR ($n = 6$). **h** Quantitative results of TUNEL staining in the heart ($n = 6$). **i** DNA fragments in the heart ($n = 6$). **j** Quantitative results of the cross-sectional area of cardiomyocyte 4 weeks post-I/R surgery ($n = 6$). **k** Quantitative results of HW/TL 4 weeks post-I/R surgery ($n = 6$). **l** Quantitative results of the collagen deposition 4 weeks post-I/R surgery ($n = 6$). **m, n** Cardiac function was presented as FS, LVIDd and LVIDs ($n = 6$). All data are expressed as the mean ± SD, and analyzed using one-way ANOVA followed by Tukey post hoc test. *$P < 0.0001$. Source data are provided as a Source Data file.

was elevated in both human IHD and mouse I/R-stressed hearts (Fig. 7b and Supplementary Fig. 14D). Yet, Tisp40 deficiency significantly decreased the mRNA and protein levels of GFPT1 in I/R-stressed hearts, but not in those without injury (Fig. 7c, d). Sequence analysis identified a highly conserved unfolded protein response element (UPRE) consensus sequence, a well-accepted Tisp40 binding site, in the GFPT1 promoter (Fig. 7e)[15]. To explore whether GFPT1 transcription could be directly regulated by Tisp40, we generated luciferase reporter plasmids containing the wild type or truncated mouse GFPT1 promoter. Co-transfection of Tisp40ΔTM-HA dramatically increased luciferase

activity in HEK293T cells expressing the wild-type GFPT1 promoter, indicating that Tisp40 directly facilitated GFPT1 transcription (Supplementary Fig. 14E). To further confirm this result, we infected Tisp40-deficient neonatal mouse cardiomyocytes with AdΔTM-HA and performed chromatin immunoprecipitation (ChIP) assay (Supplementary Fig. 14F). Semi-quantitative PCR showed enrichment of the GFPT1 promoter region in the Tisp40 precipitate (Fig. 7f). These results established occupancy by Tisp40 on the endogenous GFPT1 promoter, and we then knocked down GFPT1 to validate its necessity in Tisp40-mediated cardioprotection (Supplementary Fig. 14G). As shown in Fig. 7g and Supplementary Fig. 14H, the decreased acute cardiac I/R injury in Tisp40 cTG hearts was prevented by GFPT1 silence, as evidenced by increased IA and serum cTnT, CK-MB, and LDH levels. Meanwhile, Tisp40 overexpression-mediated inhibition on cell apoptosis and oxidative stress in I/R-stressed hearts was also blocked in those with GFPT1 silence (Fig. 7h, i and Supplementary Fig. 14I). In addition, Tisp40 overexpression failed to diminish cardiomyocyte hypertrophy and fibrosis in GFPT1-silenced hearts 4 weeks post-I/R injury (Fig. 7j–l and Supplementary Fig. 14J). Accordingly, the improved cardiac function seen in Tisp40 cTG mice after I/R surgery was abolished by GFPT1 silence, as indicated by the decreased FS and increased LVIDd and LVIDs (Fig. 7m, n). Moreover, treatment with 6-diazo-5-oxo-L-nor-leucine (DON), a pharmacological inhibitor of GFPT1, also effectively blocked the inhibitory effects on I/R-induced acute cardiac injury, remodeling, and dysfunction in Tisp40 cTG mice (Supplementary Fig. 15A–H).

Consistent with in vivo findings, we found that sI/R-induced elevations of GFPT1 mRNA and protein in NRCMs were significantly reduced by Tisp40 silence (Supplementary Fig. 16A). In contrast, NRCMs expressing an active Tisp40 displayed higher expression of GFPT1, but not GFPT2, at baseline (Supplementary Fig. 16B, C). The anti-oxidant, anti-apoptotic and cellular protective effects against sI/R insult seen in AdΔTM-HA-infected NRCMs were abolished by GFPT1 silence (Supplementary Fig. 16D–J). Meanwhile, treatment with either DON or azaserine (AZA), two independent GFPT1 inhibitors, effectively reduced the viability, but increased LDH releases in AdΔTM-HA-infected NRCMs after sI/R injury (Supplementary Fig. 16K, L). We also measured the effects of Tisp40 on the expression of other HBP-related enzymes downstream of GFPT1. As shown in Supplementary Fig. 17A, the mRNA levels of glucosamine-phosphate N-acetyltransferase (Gnpnat1) and phosphoglucomutase 3 (Pgm3), instead of UDP-N-acetylglucosamine pyrophosphorylase 1 (Uap1), Ogt or Oga, were significantly increased in Tisp40 cTG hearts at baseline. Moreover, overexpression of an active Tisp40 in NRCMs also induced the expression of Gnpnat1 and Pgm3 at baseline (Supplementary Fig. 17B). These data were consistent with previous studies identifying a conserved UPRE DNA motif in GNPNAT1 and PGM3 promoters[12, 28]. Taken together, our results suggest that Tisp40 facilitates HBP flux and cardioprotection through activating the transcription of GFPT1.

## Tisp40 is a cardiomyocyte-enriched UPR-associated transcription factor

Additional experiments were performed to determine how Tisp40 expression was regulated during cardiac I/R injury. Multiple mechanisms are implicated in the pathogenesis of cardiac I/R injury; most, if not all, of these factors are potent inducers of ER stress[29]. Tisp40 is localized to the ER, and then released to the nucleus to provoke gene transcription through binding to UPRE, suggesting that Tisp40 may work downstream of ER stress as activating transcription factor 6 (ATF6). To test this hypothesis, we first determined the interrelationship between Tisp40 expression and ER stress in NRCMs. As shown in Fig. 8a–c, treatment with thapsigargin (THA), tunicamycin (TUN) or dithiothreitol (DIT), three different ER stress inducers, significantly facilitated the expression, cleavage and nuclear accumulation of Tisp40 in NRCMs. In addition, we found that Tisp40 upregulation and nuclear translocation in sI/R-treated NRCMs or I/R-stressed hearts were dramatically inhibited by broad ER stress inhibitors, including 4-phenylbutyric acid (4-PBA) and tauroursodeoxycholic acid sodium salt (TUDCA) (Fig. 8d, e). These data reveal that I/R-induced upregulation, cleavage and nuclear accumulation of Tisp40 in the heart are mediated by ER stress. Considering the structure, subcellular localization and expression pattern of Tisp40, we define it as a cardiomyocyte-enriched UPR-associated transcription factor.

## Discussion

In this study, we demonstrate that Tisp40 expression, cleavage and nuclear accumulation in human or murine hearts and cardiomyocytes are increased by I/R injury. Gain- and loss-of-function findings identify Tisp40 as a negative regulator of I/R-induced acute cardiac injury, ventricular remodeling and dysfunction. In addition, overexpression of nuclear Tisp40 is sufficient to attenuate cardiac I/R injury in vivo and in vitro. Mechanistically, ER membrane-resident Tisp40 in I/R-injured hearts is cleaved under ER stress, and then released to the nucleus, where it directly binds to the promoter of GFPT1 and subsequently facilitates HBP flux and protein O-GlcNAcylation, thereby mitigating cardiac I/R injury (Fig. 8f). Based on these findings, we reasonably define Tisp40 as a cardiomyocyte-enriched UPR-associated transcription factor, and targeting Tisp40 may develop effective therapeutic approaches to mitigate cardiac I/R injury.

Protein homeostasis is fundamental for cellular function and organismal health, and we recently found that maintaining proteostasis by restoring proteasomal activity significantly prevented cardiac injury and remodeling in diabetic hearts[30]. At least one-third of all proteins are synthesized, modified and folded in the ER, and a stable ER environment is necessary for proteostasis and cellular function. Multiple mechanisms, including oxidative stress, are implicated in the pathogenesis of cardiac I/R injury, and most, if not all, of these factors are potential triggers to disturb ER function, eventually leading to ER stress[30]. ER stress is initially worked as an adaptive response to restore proteostasis; however, prolonged ER stress results in cell death and cardiac dysfunction. To overcome ER stress and recover ER function, UPR is activated through the stress sensors, including ATF6 and inositol-requiring protein 1α/spliced X-box-binding protein 1 (IRE1α/XBP1s)[31]. ATF6, a transmembrane ER protein, functions as a bZIP transcription factor after cleavage by S1P and S2P proteases, and facilitates protein folding through inducing the expression of ER stress response genes. Previous findings from Glembotski's laboratory revealed that ATF6 KO hearts showed increased damage and decreased function after I/R, and that transgenic mice expressing a conditionally active ATF6 in cardiomyocytes were resistant to cardiac I/R injury[32, 33]. Upon activation, IRE1α undergoes homodimerization with increased endoribonuclease activity, which subsequently causes splice of Xbp1 mRNA to generate XBP1s protein. XBP1s binds to the promoters containing ER stress response element or UPRE to maintain ER function. Consistently, we herein found that ER membrane-resident Tisp40 was upregulated and cleaved by ER stress in I/R-injured hearts or cardiomyocytes, and then translocated to the nucleus to work as a transcription factor.

Protein function is regulated at multiple levels, including post-translational modifications. O-GlcNAcylation, one of the most common post-translational modifications of nuclear, cytoplasmic and mitochondrial proteins, is identified as an auto-protective alarm or stress response to prevent cell death. Unlike protein phosphorylation involving numerous kinases, only two enzymes (OGA and OGT) are involved in the addition and removal of one GlcNAc molecule from the Ser/Thr amino acid residues of target proteins, making the regulation more quickly and accurate[5]. GFPT is the rate-limiting enzyme for the HBP, and converts fructose-6-phosphate and glutamine to produce GlcN-6P, thereby facilitating HBP flux. Eukaryotic GFPT consists of two isoforms, GFPT1 (ubiquitously expressed in the pancreas, placenta,

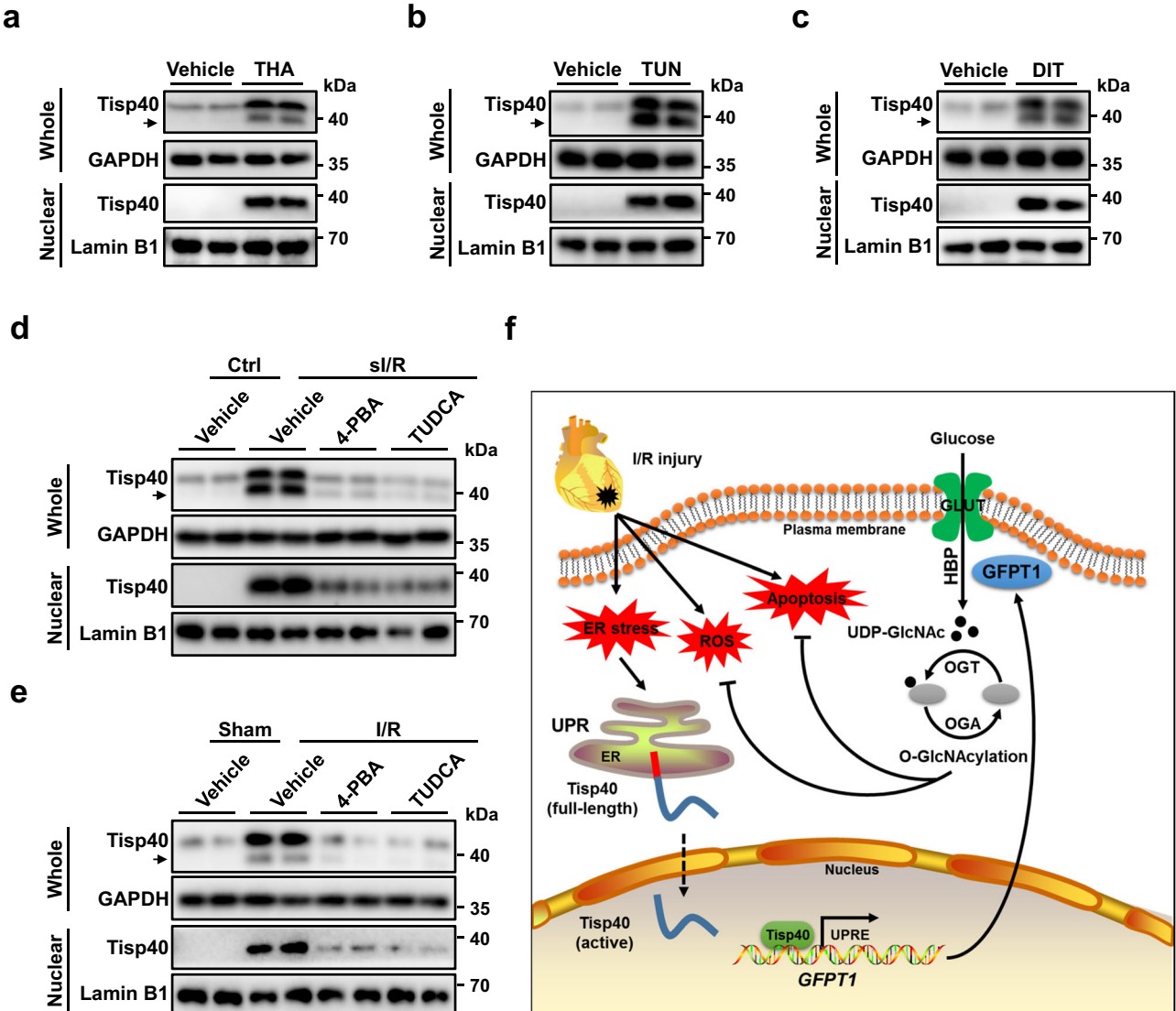

**Fig. 8 | Tisp40 is a cardiomyocyte-enriched UPR-associated transcription factor. a–c** NRCMs were stimulated with thapsigargin (THA), tunicamycin (TUN) or dithiothreitol (DIT) for 6 h, and then whole-cell lysates and nuclear lysates from NRCMs were prepared for western blot ($n = 6$). **d** To suppress ER stress in NRCMs, 4-phenylbutyric acid (4-PBA) and tauroursodeoxycholic acid sodium salt (TUDCA) were added during sI/R injury. Next, whole-cell lysates and nuclear lysates from NRCMs were prepared for western blot ($n = 6$). **e** To inhibit ER stress in mice, 4-PBA or TUDCA was administered by a single intraperitoneal injection 15 min before reperfusion, and then whole-cell lysates together with nuclear lysates from the heart were prepared for western blot ($n = 6$). **f** Schematic diagram of the molecular mechanisms underlying Tisp40-regulated cardiac I/R injury. ER membrane-resident Tisp40 in I/R-injured hearts is upregulated and cleaved under ER stress, and then released to the nucleus, where it directly binds to the promoter of GFPT1 and subsequently facilitates HBP flux and protein O-GlcNAcylation, thereby mitigating cardiac I/R injury.

testis, liver, skeletal muscle, and heart) and GFPT2 (primarily expressed in the heart, brain, and spinal cord), which show 75% amino acid sequence identity in mice and humans[6]. Previous studies have revealed that GFPT2 is the main type of GFPT in the heart[34]. Accordingly, Belke et al. found that GFPT2 downregulation resulted in decreased protein O-GlcNAcylation[35]. Ishikita et al. demonstrated that GFPT2 silence inhibited HBP/O-GlcNAcylation, and subsequently prevented pathological cardiac hypertrophy[34]. In contrast, findings from Nabeebaccus et al. indicated that GFPT2 was only identified in cardiac fibroblasts, whereas GFPT1 was expressed in both myocytes and fibroblasts in the heart, and that protein O-GlcNAcylation in the heart was regulated by GFPT1 not GFPT2[36]. Consistently, Tran et al. revealed that GFPT1 overexpression was sufficient to provoke HBP flux and subsequently caused cardiac remodeling following chronic stimulation[9]. Interestingly, we herein found that Tisp40, a cardiomyocyte-enriched ER membrane-resident transmembrane protein, sensed ER stress during

cardiac I/R injury and then translocated to the nucleus after cleavage to bind to a conserved UPRE of the GFPT1 promoter, thereby facilitating HBP flux and O-GlcNAc protein modifications. Treatment with broad ER stress inhibitors (e.g., 4-PBA and TUDCA) dramatically inhibited I/R-induced upregulation, cleavage and nuclear accumulation of Tisp40 in the heart. Based on these findings, we reasonably identify Tisp40 as a cardiomyocyte-enriched UPR-associated transcription factor.

Contrary to our findings, Qin and colleagues found that global Tisp40 deficiency significantly prevented I/R-induced tubular apoptosis, fibrosis and renal dysfunction[37, 38]. Mechanistically, Tisp40 aggravated renal I/R injury through activating NF-κB-mediated inflammation and pyroptosis[26, 39]. Similarly, Stelzer et al. previously revealed that Tisp40 bound specifically to a NF-κB-binding element[25]. Yet, NF-κB activation and cardiac inflammation in I/R-stressed hearts were unaffected by Tisp40 deficiency in our study. Myeloid cells are the main inflammatory cells in the heart, and Tisp40 was also modestly

upregulated in cardiac macrophages after I/R surgery. By performing bone marrow transplantation study, we found that macrophage Tisp40 was dispensable for cardioprotection against I/R injury. It is well-accepted that molecules expressed by different cardiac cells may differentially modulate global cardiac function[40]. Therefore, the discrepancy in the heart and kidney might be attributed to the cellular origin of Tisp40.

In summary, we herein define Tisp40 as a cardiomyocyte-enriched UPR-associated transcription factor, and provide proof of concept that targeting Tisp40 can develop effective therapeutic approaches to mitigate cardiac I/R injury.

# Methods

## Ethics statement
All the experimental procedures were approved by the Animal Care and Use Committee of Renmin Hospital of Wuhan University (approval No. 20191106), and were also in accordance with the *Guidelines for the Care and Use of Laboratory Animals* published by the US National Institutes of Health. All experimental procedures involving human samples in this study were in accordance with the Declaration of Helsinki and also approved by the Review Board of Renmin Hospital of Wuhan University. Written informed consent was obtained from all patients and donors.

## Reagents
Evans blue (#E2129), 2,3,5-triphenyltetrazolium chloride (TTC, #T8877), 4-PBA (#P21005), TUDCA (#T0266), hydroxyproline assay kit (#MAK008), lucigenin (#M8010), GlcN (#G4875), GlcNAc (#A8625), DON (#D2141), AZA (#A4142), ALX (#A7413), TMG (#SML0244) and O-(2-acetamido-2-deoxy-D-glucopyranosylidenamino) N-phenylcarbamate (PUGNAc, #A7229) were purchased from Sigma-Aldrich (St. Louis, MO, USA). Glucosamine-6-phosphate (GlcNAc-6P, #Y63951) and UDP-GlcNAc (#S27600) were purchased from Shanghai yuanye Bio-Technology Co., Ltd. (Shanghai, China). Dihydroethidium (DHE, #KGAF019) was obtained from Jiangsu KeyGEN BioTECH Corp., Ltd (Jiangsu, China). Cell counting kit-8 (CCK-8, #C0037), Lipo6000™ transfection reagent (#C0526) and 2′,7′-dichlorodihydrofluorescein diacetate (DCFH-DA, #S0033) were purchased from Beyotime (Shanghai, China). LDH assay kit (#ab102526) and 3-NT ELISA assay kit (#ab116691) were obtained from Abcam (Cambridge, UK). THA (#sc-24017), TUN (#sc-3506) and DIT (#sc-29089) were obtained from Santa Cruz Biotechnology (Dallas, Texas, USA). MDA assay kit (#A003) and 4-HNE ELISA kit (#H268) were purchased from Nanjing Jiancheng Bioengineering Institute (Nanjing, China). Cell Death Detection ELISA^PLUS kit (#11774425001) was purchased from Roche Applied Science (Basel, Switzerland). Insulin mouse ELISA kit (#EMINS), EnzChek caspase3 assay kit (#E13183), Amplex Red Hydrogen Peroxide/Peroxidase assay kit (#A22188), Pierce™ BCA protein assay kit (#23225), NE-PER nuclear and cytoplasmic extraction reagent (#78833), IL-6 mouse ELISA kit (#KMC0061), TNF-α mouse ELISA kit (#BMS607-3), the goat anti-mouse IgG Alexa Fluor 488 secondary antibodies (#A11001), goat anti-rabbit IgG Alexa Fluor 568 secondary antibodies (#A11011), Alexa Fluor™ 488 conjugated Wheat Germ Agglutinin (WGA, #W11261) and SlowFade™ gold antifade reagent with DAPI (#S36939) were purchased from Invitrogen (Carlsbad, CA, USA). GTVisionTM+ Detection System/Mo&Rb reagent (#GK600710A) was obtained from Gene Tech Company Limited (Shanghai, China). Apop-Tag Plus In Situ Apoptosis Fluorescein Detection Kit (#S7111) was purchased from Millipore (Billerica, MA, USA). Adenoviral vectors carrying a HA-tagged truncated mouse Tisp40 lacking the TM domain (amino acids 1-269, accession no. NP_084356; AdΔTM-HA) or a scramble control (AdCtrl) were generated by DesignGene Biotechnology (Shanghai, China), and the core sequences were also cloned into cardiotropic adeno-associated virus serotype 9 (AAV9) vectors under a cTnT promoter to construct AAV9ΔTM-HA or AAV9Ctrl. Two independent short hairpin RNAs (#TL700777V) against rat Tisp40 (shTisp40

shTisp40#) or a control shRNA carried by the lentiviral vectors were obtained from OriGene Technologies, Inc. (Rockville, MD, USA). AAV9 vectors carrying a short hairpin RNA against mouse glutamine-fructose-6-phosphate transaminase 1 (shGfpt1) or shRNA under the cTnT promoter were constructed to silence cardiac GFPT1 expression in vivo. Small interfering RNA against rat GFPT1 (siGfpt1, #SR510107), rat OGT (siOgt, #SR503126), rat OGA (siOga, #SR511655) or scramble siRNA were all obtained from OriGene Technologies, Inc.

## Animals and treatments
Global Tisp40 KO mice (stock No. RBRC01942) were purchased from the RIKEN BioResource Research Center and backcrossed to C57BL/6 strain over ten generations. Exons 5-10 encoding bZIP and TM domains of Tisp40 were replaced with a neomycin cassette in these Tisp40 KO mice[19]. Heterozygous Tisp40-deficient mice were crossed with each other to generate homozygous Tisp40 KO mice, and the WT littermates were used as the negative control. Genotyping was performed using the following primers: primer 1: 5′-CTAAAGCGCATGCTCCAGACTGCC-3′; primer 2: 5′-GGGCCAGTCTCCTACCCAAGC-3′; primer 3: 5′-TTCTGTTGCATAGGCCTGATGACC-3′. To establish Tisp40 cTG mice, full-length mouse Tisp40 cDNA was cloned under control of the cardiac α-MHC promoter, which was then microinjected into fertilized mouse embryos[41, 42]. Four independent transgenic lines were constructed, and the matched non-transgenic (NTG) littermates were selected as the control. All mice were born at the expected Mendelian ratio, bred normally, displayed normal behaviors and survival rates as previously observed[18, 19]. These mice were kept in Individually Ventilated Cages with the density of 4−6 mice per cage under a specific pathogen-free, environmentally suitable barrier system (20−25 °C and 45−55% humidity) on a regular 12 h light/dark cycle at the Cardiovascular Research Institute of Wuhan University. All mice were fed with a irradiated chow diet (#1035 for reproductive feeding and #1025 for maintenance feeding, Beijing HFK Bioscience Co., Ltd, Beijing, China), with free access to drinking water.

To induce cardiac I/R injury in vivo, male mice aged 10−12 weeks were firstly anesthetized by intraperitoneal injections of 3% pentobarbital sodium, and the body temperature was maintained at $37 \pm 1$ °C by a warming pad. Next, the heart was exposed by a left thoracotomy, and the left anterior descending (LAD) artery 1 mm below the tip of the left atrial appendage was ligated with a 7-0 silk suture over a piece of PE-10 tubing. Ischemia was confirmed by both ST elevation on surface electrocardiogram and visual blanching. After occlusion for 45 min, the tubing was removed, and the ligature was released to achieve reperfusion. Serum and heart samples were collected at the indicated time points after reperfusion. Sham-operated mice received same surgical procedures without ligating the LAD artery. In addition, 4-PBA (100 mg/kg) or TUDCA (250 mg/kg) was administered by a single intraperitoneal injection 15 min before reperfusion to inhibit ER stress[43, 44]. To specifically overexpress nuclear Tisp40 in the heart, Tisp40 KO mice received a single intravenous injection of AAV9ΔTM-HA at a dose of $1 \times 10^{11}$ viral genomes per mouse 4 weeks prior to I/R surgery[45, 46]. To knock down endogenous GFPT1 in the heart, Tisp40 cTG mice were injected from the tail vein with $1 \times 10^{11}$ viral genomes of shGfpt1 4 weeks before I/R surgery. In addition, Tisp40 cTG mice were also intraperitoneally injected with DON (1 mg/kg) every other day for 20 consecutive days prior to I/R surgery to suppress GFPT1[27]. For OGA inhibition in vivo, Tisp40 KO mice were intraperitoneally injected with NButGT (50 mg/kg) daily for 14 consecutive days or TMG (20 mg/kg) every other day for 20 consecutive days. And the last injections of NButGT or TMG were done 30 min before surgery[27, 47]. All mice were sacrificed at indicating times by cervical dislocation.

## Bone marrow transplantation
Bone marrow transplantation was performed to investigate the role of macrophage Tisp40 in cardiac I/R injury[48, 49]. Briefly, 4−6-week-old

male Tisp40 KO mice were lethally irradiated with a total dose of 1300 Rads (650 Rads per time, 4 h interval). Bone marrow cells isolated from the femurs of Tisp40 KO or WT mice were injected into the tail vein of recipient mice ($1 \times 10^7$ cells per mouse) 4 h after the last irradiation. These chimeric mice were maintained on antibiotic-containing water (100 mg/L levofloxacin and 100 mg/L fluconazole) for 2 weeks and then were kept for an additional 2 weeks before I/R surgery. Peripheral blood was collected for the PCR analysis of hematologic chimerism using the following primers: primer 1: 5′-GGGCCAGTCTCCTACC-CAAGC-3′; primer 2: 5′-TTCTGTTGCATAGGCCTGATGACC-3′.

## Transthoracic echocardiography

Cardiac contractile function was measured by transthoracic echo-cardiography in conscious, gently restrained mice at baseline, 2 and 4 weeks post-I/R surgery, using a Vevo® 3100 high-resolution Pre-clinical Imaging System (FUJIFILM VisualSonics, Toronto, Canada) equipped with a 30-MHz linear ultrasound transducer (MX 400) as we recently described[50]. Briefly, mice were lightly anesthetized by 1.5% isoflurane, and then two-dimensional guided M model images crossing the anterior/posterior wall of the left ventricle were recorded to measure the LVIDd, LVIDs, IVSd, IVSs, and HR from at least five consecutive cardiac cycles. FS was calculated as (LVIDd − LVIDs)/LVIDd.

## Histological analysis

To evaluate infarct size of the heart, mice were re-anesthetized, and the chest was opened to re-ligate the LAD artery at the same location following 24 h reperfusion. Next, 2% Evans blue dye was injected into the jugular vein to demarcate the ischemic area at risk (AAR) and remote area, and then the heart was excised, frozen at −20 °C for 20 min and sectioned into 4–5-μm slices that were subsequently incubated with 1% TTC solution at 37 °C for an additional 15 min to delineate the IA. The IA (pale), AAR (not blue) and total LV area were quantified and analyzed by Image-Pro Plus 6.0 software (Media Cybernetics, Bethesda, MD, USA) in a blinded manner. To evaluate whole morphology, heart samples were excised 4 weeks post-I/R surgery, fixed in 4% neutral formaldehyde for 48 h, dehydrated, embedded in paraffin and sectioned to 5 μm slices. Next, the cardiac slices were incubated with WGA working buffer (1:200) at 37 °C for 1 h to examine the cross-sectional area of cardiomyocytes, and more than 200 cells from 6 mice per group were included. To measure collagen deposition, picrosirius red (PSR) staining was performed according to the standard protocols[30, 51, 52].

## Cells and treatments

NRCMs were isolated from the left ventricles of 1–3-day-old Sprague−Dawley rats using enzymatic digestion, and bromodeoxyuridine (100 μmol/L) was used to remove the residual cardiac fibroblasts[30, 53]. The purity of cardiomyocytes was confirmed over 90% by an immunostaining for α-actinin. For sI/R injury in vitro, cardiomyocytes were washed twice with phosphate-buffered saline (PBS) and incubated with the Esumi ischemic buffer under a hypoxic condition (95% $N_2$, 5% $CO_2$) for 4 h, which were then washed twice with PBS and incubated with fresh DMEM/F12 containing 10% fetal bovine serum (FBS) under a normoxic condition (95% air, 5% $CO_2$) overnight. NRCMs incubated in fresh control buffer under normoxia were used as the Ctrl[12, 54]. To induce ER stress, NRCMs were stimulated with THA (1 μmol/L), TUN (0.5 μg/mL) or DIT (0.5 mmol/L) for 6 h[12, 55]. 4-PBA (0.5 mmol/L) and TUDCA (0.5 mmol/L) were used to suppress ER stress in NRCMs during sI/R injury[44, 56]. To overexpress nuclear Tisp40, NRCMs were pre-infected with AdΔTM-HA at a multiplicity of infection (MOI) of 20 for 4 h, and then incubated with fresh DMEM/F12 containing 10% FBS for an additional 48 h before sI/R stimulation. For Tisp40 silence, NRCMs were pre-infected with shTisp40 or shTisp40# at a MOI of 30 for 4 h. To knock down endogenous GFPT1, OGT and OGA in vitro, NRCMs were pre-transfected with 50 nmol/L siGfpt1,

siOgt or siOga for 4 h using a Lipo6000™ transfection reagent according to the manufacturer's instructions, followed by the incubation in fresh medium for an additional 24 h before Tisp40 manipulation. Meanwhile, DON (20 μmol/L), AZA (5 μmol/L), ALX (2.5 mmol/L), TMG (10 μmol/L) or PUGNAc (200 μmol/L) were added to the medium during reperfusion to inhibit the activities of GFPT1, OGT or OGA, respectively[9, 47, 57]. Furthermore, either GlcN (50 μmol/L) or GlcNAc (10 mmol/L) was added to the medium of Tisp40-deficient NRCMs to activate hexosamine biosynthesis during reperfusion[9, 12]. To enhance the clinical impact and translational value of our study, hiPSC-CMs were prepared[58]. Briefly, hiPSCs were cultured in mTeSR™1 medium on LDEV-free Corning® Matrigel® hESC-Qualified Matrix (Corning, NY, USA)-coated six-well plates, and then, $3.5 \times 10^5$ single hiPSCs were prepared and seeded onto pre-coated 12-well plates with 10 μmol/L Y-27632. Next, the medium was replaced with fresh mTeSR™1 without Y-27632, and hiPSCs were cultured for an additional 24 h. After grown to 95% confluency, hiPSCs were cultured according to the manufacturer's instructions with STEMdiff™ Cardiomyocyte Differentiation Medium containing Supplement A, B, or C, respectively. On the 8th day, hiPSCs were cultured with STEMdiff™ Cardiomyocyte Maintenance Medium, and the medium was replaced every 2 days. On the 15th day, these hiPSCs were used as hiPSC-CMs and subjected to sI/R in vitro. To clarify the expression of Tisp40, macrophages, cardiomyocytes, and cardiac fibroblast in adult hearts were isolated[49, 59].

## Immunofluorescence staining

Immunofluorescence staining was performed on heart and cell samples[60, 61]. For immunofluorescence staining of cardiac slices, paraffin-embedded sections underwent deparaffinization, rehydration and antigen retrieval with citric acid buffer (pH = 6.0). For immunofluorescence staining of cell coverslips, cells were fixed with 4% paraformaldehyde for 15 min and permeabilized in 1% Triton X-100 for 5 min at room temperature. After blocking the non-specific binding with 10% goat serum, cardiac slices or cell coverslips were incubated with the primary antibodies (Supplementary Table 1) at 4 °C overnight, and stained with the goat anti-mouse IgG Alexa Fluor 488 or goat anti-rabbit IgG Alexa Fluor 568 secondary antibodies (1:200 dilution) at 37 °C for an additional 1 h. The nuclei were visualized with SlowFade™ gold antifade reagent with DAPI, and immunofluorescent images were obtained by a DP74 fluorescence microscope (OLYMPUS, Tokyo, Japan).

## Immunohistochemical staining

Immunohistochemical staining was performed to analyze the expression of α-SMA in the heart[60, 62]. Briefly, deparaffinized sections were subjected to a high-pressure antigen retrieval process, and then blocked with 3% hydrogen peroxide and 10% goat serum. Next, cardiac slices were incubated with anti-α-SMA (Supplementary Table 1) at 4 °C overnight and horseradish peroxidase (HRP)-conjugated secondary antibodies at 37 °C for an additional 1 h, visualized with diaminobenzidine and analyzed using the Image-Pro Plus 6.0 software.

## Western blot

Total proteins were extracted using RIPA lysis buffer containing a protease inhibitor cocktail and phosphatase inhibitors, and the concentration was quantified using a commercial kit[59, 63]. An equal amount of proteins were separated by sodium dodecyl sulfate-polyacrylamide gel electrophoresis, transferred to polyvinylidene fluoride membranes, blocked with 5% skim milk and incubated with the primary antibodies (Supplementary Table 1) at 4 °C overnight. On the next day, the membranes were probed with HRP-conjugated secondary antibodies at room temperature for 1 h and visualized using ChemiDoc™ XRS+ System (Bio-Rad Laboratories, Inc.) with the electrochemiluminescence reagent. Protein bands were analyzed using Image

Lab Software (version 6.0, Bio-Rad Laboratories, Inc.) and normalized to internal controls. For O-GlcNAc analysis, tissues or cells were lysed in the presence of PUGNAc (40 µmol/L)[12]. Nuclear extracts were prepared by the NE-PER nuclear and cytoplasmic extraction reagent according to the manufacturer's instructions, and lamin B1 was used as the internal control.

## Quantitative real-time PCR

Total RNA was extracted using TRIzol reagent, and cDNA synthesis was performed with a Transcriptor First Strand cDNA Synthesis Kit (Roche, Basel, Switzerland). Quantitative real-time PCR was performed using SYBR Green I Master Mix (Roche) on the Roche LightCycler 480 system[30, 59]. Gene expression was calculated using the $2^{-\Delta\Delta Ct}$ method, and primer sets were shown in Supplementary Table 2.

## Collagen content analysis

Total collagen content was measured by evaluating the level of hydroxyproline, a major component of collagen[64]. Briefly, fresh left ventricles were homogenized and hydrolyzed in hydrochloric acid (12 mol/L) at 120 °C for 3 h, followed by a dryness under vacuum at 60 °C. Next, the samples were dissolved, vortexed and incubated with Chloramine T/Oxidation Buffer Mixture at room temperature for 5 min, and then incubated with diluted 4-(dimethylamino) benzaldehyde reagent at 60 °C for 1.5 h. The absorbance was measured at 560 nm using a BioTek microplate reader (Winooski, Vermont, USA), and the hydroxyproline content was calculated by normalizing to tissue weight. To determine the content of soluble collagen, fresh tissues were solubilized in pepsin-acid buffer and centrifuged, with the supernatants incubated with a trichloroacetic acid solution to precipitate the soluble collagen. The soluble collagen was then hydrolyzed in hydrochloric acid and subjected to hydroxyproline content measurements, while the insoluble collagen content was determined by subtracting the amount of soluble collagen from total collagen.

## DHE, DCFH-DA, and TUNEL staining

Cardiac $O_2^-$ level was determined using DHE staining[45, 65]. Briefly, fresh frozen cardiac slices were incubated with DHE solution (5 µmol/L) at 37 °C for 30 min in the dark, and the fluorescent intensity was measured by a DP74 fluorescence microscope. DCFH-DA staining was performed to evaluate intracellular ROS levels in vitro[65, 66]. Briefly, cells were incubated with DCFH-DA solution (5 µmol/L) at 37 °C for 30 min in the dark, and then, the images were recorded with fluorescence microscope after washing the cells for three times with serum-free medium. Cell apoptosis in vivo and in vitro was evaluated by TUNEL staining with a commercial kit according to the manufacturer's instructions, and immunofluorescence staining with an anti-α-actinin antibody was performed to label cardiomyocytes[30, 63]. The apoptotic index was expressed as the percentage of apoptotic α-actinin positive cardiomyocytes over the total cardiomyocytes.

## Biochemical analysis

Serum levels of cTnT, CK-MB, LDH, triglyceride and total cholesterol were measured using an ADVIA® 2400 automatic biochemical analyzer (Siemens Healthcare Diagnostics, Tarrytown, NY, USA)[45, 46]. Fasting blood glucose was examined using a One Touch Ultra Easy glucometer (Life Scan, Wayne, PA, USA), whereas serum insulin levels were determined with a commercial kit according to the manufacturer's instructions. The HOMA-IR index was calculated as fasting blood glucose × serum insulin/22.5[65]. In addition, fresh heart homogenates or cell lysates were prepared, and the levels of MDA, 4-HNE, and 3-NT were measured using the commercial kits[45, 46]. To assess cardiac $H_2O_2$ level, fresh left ventricular blocks were incubated with Amplex Red reagent (100 µmol/L) and HRP (1 U/mL) at 37 °C for 30 min in the dark, and the absorbance was measured at 560 nm to evaluate cardiac $H_2O_2$ level. $O_2^-$ production in the heart was measured according to the following protocols[67]. Briefly, fresh left ventricles were homogenized and incubated with lucigenin (5 µmol/L), NADH or NADPH (500 µmol/L). Next, lucigenin-enhanced chemiluminescence was continuously monitored at 37 °C every 3 s, and the levels of $O_2^-$ were calculated as the difference with or without superoxide dismutase (100 µg/mL). Cell viability was evaluated using CCK-8 method according to the manufacturer's instructions[53, 66]. DNA fragments were determined using a Cell Death Detection ELISA[PLUS] kit according to the manufacturer's instructions. And caspase3 activities were measured by detecting the fluorogenic change of Z-DEVD-AMC using a commercial kit[30, 65]. LDH releases were detected to further evaluate cardiomyocyte injury in vitro, and calculated as (LDH level in ischemia medium + LDH in reperfusion medium)/(LDH in ischemia medium + LDH in reperfusion medium + LDH in cell lysate)[12].

## Metabolites analysis

Levels of GlcNAc-6P and UDP-GlcNAc in the heart were measured using targeted ultra-high performance liquid chromatography-tandem mass spectrometry (UHPLC-MS/MS). Briefly, fresh heart samples were vortexed for 30 s, homogenized in extraction solution (acetonitrile: methanol: water = 2:2:1) for 4 min, and sonicated for 5 min on ice, which were then centrifuged at 4 °C for 15 min to obtain the supernatants. Next, the supernatants or standard samples were separated by a Waters Premier Amide (2.1 mm × 100 mm, 1.7 µm) on an Agilent 1290 Infinity II series UHPLC System (Agilent Technologies, Palo Alto, CA, USA) using 0.1% ammonia and 10 mmol/L ammonium formate (phase A), and acetonitrile (phase B) as the mobile phases (300 µL/min). The metabolites were detected using an Agilent 6495 Triple Quadrupole Mass Spectrometer (Agilent Technologies), equipped with an AJS electrospray ionization interface, and then normalized to tissue weight. Typical ion source parameters were: capillary voltage = +3000 V, nozzle voltage = +1500 V, gas ($N_2$) temperature = 250 °C, gas ($N_2$) flow = 11 L/min, sheath gas ($N_2$) temperature = 400 °C, sheath gas flow = 12 L/min, nebulizer = 35 psi.

## Bioinformatics analysis

To clarify the potential downstream target of Tisp40, we retrieved a microarray hybridization dataset from the online Gene Expression Omnibus database (https://www.ncbi.nlm.nih.gov/geo/query/acc.cgi?acc=GSE7223). The raw read data were derived from human LNCaP cells expressing an active Tisp40 upon RSL1 treatment, based on Affymetrix Human Genome U133 Plus 2.0 arrays[24]. These probe sets were integrated, corrected, logarithmic transformed and filtered to screen the differentially expressed genes (DEGs). Statistically significant of DEGs from the pairwise comparisons were defined as Fold Change >2 and an adjusted P value < 0.05. KEGG pathway enrichment analysis was performed to determine the significant functions and pathways of DEGs.

## ChIP assay

For ChIP assay, neonatal cardiomyocytes were isolated from Tisp40 KO mice and infected with AdΔTM-HA at a MOI of 20 for 4 h, which were then cultured in fresh medium for an additional 48 h. Afterward, these cells were cross-linked with 1% formaldehyde at room temperature for 10 min and then stopped by incubating with glycine (125 mmol/L) for an additional 5 min. Next, cells were lysed with the nuclear extracts prepared, sonicated and incubated with protein G agarose beads-conjugated anti-HA (Supplementary Table 1) or anti-IgG isotype control at 4 °C overnight. After pull-down, immunocomplexes were washed five times and processed by reverse cross-linking, proteinase K digestion and DNA precipitation. PCR was performed using primers spanning the Tisp40 binding site in mouse GFPT1 promoter (Supplementary Table 2). Genomic DNA before immunoprecipitation was used as an input control.

### Luciferase reporter assay

The wild-type mouse GFPT1 promoter was cloned into a pGL3-basic vector to construct the GFPT1-luc plasmid, and a truncated GFPT1 promoter lacking the UPRE region was also generated. These two constructs were co-transfected into HEK293T cells (#GNHu17, Type Culture Collection of the Chinese Academy of Sciences, Shanghai, China) together with a mouse Tisp40ΔTM-HA plasmid. After 48 h, cells were lysed to detect firefly and Renilla luciferase activity with a Dual-Luciferase reporter Assay system (Promega) according to the manufacturer's instructions.

### Human heart samples

Human heart samples were obtained from the left ventricular free wall or apex of IHD patients undergoing heart transplantation, and the control samples were obtained from donor hearts that were unsuitable for transplantation for non-cardiac diseases[59, 68].

### Statistical analysis

All data are expressed as the mean ± standard deviation (S.D.), and analyzed using GraphPad Prism (version 7.0). Comparisons between two groups with a normal distribution and homogeneity of variance were performed using an unpaired two-tailed Student's $t$ test, whereas one-way analysis of variance (ANOVA) followed by Tukey post hoc test was conducted for comparisons among three or more groups. Differences between groups over time were evaluated by repeated measures ANOVA. A $P$ value less than 0.05 was considered statistically significant.

### Reporting summary

Further information on research design is available in the Nature Portfolio Reporting Summary linked to this article.

## Data availability

The data that support the findings of this study are available within the main text, its Supplementary Information file, and Source data. The microarray hybridization dataset is downloaded from the online Gene Expression Omnibus database (https://www.ncbi.nlm.nih.gov/geo/query/acc.cgi?acc=GSE7223)[24]. Source data are provided with this paper.

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

## Acknowledgements

This work was supported by grants from The Regional Innovation and Development Joint Fund of National Natural Science Foundation of China (No. U22A20269), the Fundamental Research Funds for the Central Universities (No. 2042023kf0046), the Key Project of the National Natural Science Foundation (No. 81530012) and Science and Technology Planning Projects of Wuhan (No. 2018061005132295).

## Author contributions

X.Z., C.H. and Q.Z.T. conceived the hypothesis. X.Z., C.H., Z.G.M., M.H., X.P.Y., and C.Y.K. performed the experiments. X.Z., C.H., Y.P.Y., C.Y.K., S.S.W., and T.T. analyzed the data. X.Z. drafted the manuscript. X.Z., C.H., and Q.Z.T. revised the manuscript. Q.Z.T. supervised the project.

## Competing interests

The authors declare no competing interests.
