## [Peer review file · Nature Communications]

REVIEWER COMMENTS

Reviewer #1 (Remarks to the Author):

The hexosamine biosynthetic pathway (HBP) is a well-established cytoprotective pathway. Here, the manuscript addressed an ER-resident protein, Tisp40. In other cell types, Tisp40 has been shown to be protective. The present manuscript contains an extraordinary amount of data, which seems carefully performed. The approaches included a battery of gain- and loss-of-function approaches to address the central goal of the study. In short, these data provide an exhaustive amount of convergent evidence that increasing Tisp40 is protective, while deleting/inhibiting Tisp40 is injurious, following various in vivo and in vitro models of ischemia/reperfusion and hypoxia/reoxygenation injury. Despite the extraordinary volume of data presented—which, perhaps, could be streamlined—the observations remain clear, and the conclusions are supported by the data. The authors should consider the following:

General:

1. The rationale for the study is a bit difficult to understand. The motivation for studying HBP/O-GlcNAc is established by many prior studies. As stated, the authors were interested in targeting this, so, they addressed an endoplasmic reticulum resident protein, Tisp40. Tisp40 was known to promote survival of other cell types, but its role in the heart is not known. If targeting O-GlcNAcylation was the goal, perhaps more translational approaches could have been used and/or identification of new mechanisms. Either way, the introduction to the manuscript does little to prepare the reader for the somewhat abrupt linkage between O-GlcNAc and Tisp40.

Specific:

1. Lines 993–994 do not apply to Figure 8.

Reviewer #2 (Remarks to the Author):

In this study, Dr. Tang and colleagues aimed to characterize the role of Tisp40, an understudied protein in ER stress, cardio-protection, and hexosamine biosynthesis. The authors identified Tisp40 as an increased transcription factor by myocardial ischemia/reperfusion (I/R). In vivo tests demonstrated that Tisp40 protected the heart against I/R injury. At the mechanistic level, Tisp40 targeted GFPT1, the rate-limiting enzyme of hexosamine biosynthesis. Additional loss-of-function experiments revealed that

GFPT1 mediated, at least partially, the protective effect of Tisp40 under I/R. Overall, the study was thoughtfully designed, and the conclusions are largely supported by the findings.

1. Figure 2, the IA in the global KO mice is quite high. I suspect that there will be mortality issue. Can the authors provide survival rate comparing WT and KO mice after I/R?
2. Figure 2D, Figure 4B, Figure 4E, since there is no co-staining with cardiomyocyte marker, it is important to emphasize that the positive signal may not be cardiomyocytes. For example, they might be immune cells.
3. The Y axis of all data should start from 0, not a random number. This issue is pertinent to Figures 3E, 3F, 3B, 4K, 5K, 6I, 7G, 7K, 7M, 7N, S2C, S6G, S6H, S7G, S8K, S9J, S12G, S15A, and S15H.
4. It is interesting to know if knockout of Tisp40 affects the expression of other ER stress signaling transducers, such as IRE1a, XBP1s, ATF6, PERK, ATF4, and GRP78.
5. Throughout the manuscript, the authors used H7E staining images to quantify cardiomyocyte size. This is not accurate. WGA staining is preferred. Moreover, the N number refers to the number of mice used or the number of cardiomyocytes? It is not clear. Along the same line, in addition to fibrosis as quantified by the authors, cardiac pathological remodeling can be assessed by gene expression of the fetal gene program, including NPPA, NPPB, beta-MHC, etc.
6. Does Tisp40 affect the transcription of other enzymes of the hexosamine biosynthetic pathway, such as GNPAT1, UAP1, etc?

Reviewer #3 (Remarks to the Author):

Zhang et al TSP40 NCOMMS-23-00930 " Tisp40, a novel cardiomyocyte-enriched UPR-associated transcription factor, prevents cardiac ischemia/reperfusion injury through the hexosamine biosynthetic pathway "

This manuscript describes studies done mostly in a mouse model of ischemia/reperfusion injury (IRI) and testing how Tisp40, also known as Creb3l1, affects reperfusion injury. The study is clearly described and the results are well depicted and appropriately interpreted.

Comments:

1- The authors should refrain from concluding that I/R or ER stress causes the cleavage of the ER form of Tisp40 and the translocation of the product to the nucleus. They showed that there are two forms of Tisp40, one larger than the other, and that the smaller of the forms concentrates in the nucleus after stress, but this does not prove their cleavage/translocation theory.

2- Related to my previous comment, it is premature to show a conclusion diagram such as in Fig. 8F, since much of the processes shown are hypothetical. This panel should be removed.

3- The Tisp40 knock out mice appear to have this gene deleted globally. A more suitable model would have been to use cardiac specific knock out, which the authors did use later for ectopic expression of Tisp40. However, given all of the other evidence the authors have provided, this is potential excusable. The authors should say in the Results that the Tisp40 knock out is global.

4- The authors should remove the word “novel” wherever it appears in the manuscript. Tisp40 is not novel, it has been previously studied, albeit in other cell/tissue types.

Reviewer #1 (Remarks to the Author):

The hexosamine biosynthetic pathway (HBP) is a well-established cytoprotective pathway. Here, the manuscript addressed an ER-resident protein, Tisp40. In other cell types, Tisp40 has been shown to be protective. The present manuscript contains an extraordinary amount of data, which seems carefully performed. The approaches included a battery of gain- and loss-of-function approaches to address the central goal of the study. In short, these data provide an exhaustive amount of convergent evidence that increasing Tisp40 is protective, while deleting/inhibiting Tisp40 is injurious, following various in vivo and in vitro models of ischemia/reperfusion and hypoxia/reoxygenation injury. Despite the extraordinary volume of data presented-which, perhaps, could be streamlined-the observations remain clear, and the conclusions are supported by the data. The authors should consider the following:

General:

1. The rationale for the study is a bit difficult to understand. The motivation for studying HBP/O-GlcNAc is established by many prior studies. As stated, the authors were interested in targeting this, so, they addressed an endoplasmic reticulum resident protein, Tisp40. Tisp40 was known to promote survival of other cell types, but its role in the heart is not known. If targeting O-GlcNAcylation was the goal, perhaps more translational approaches could have been used and/or identification of new mechanisms. Either way, the introduction to the manuscript does little to prepare the reader for the somewhat abrupt linkage between O-GlcNAc and Tisp40.

Response: Thanks for your comment. Tisp40 is an endoplasmic reticulum (ER) membrane-resident type II transmembrane protein and works as a transcription factor after cleavage by a “regulated intramembrane proteolysis” mechanism. Previous studies have linked Tisp40 with cell survival, a key mechanism in cardiac I/R injury. Therefore, we investigated the role and molecular basis of Tisp40 in cardiac I/R injury in this study. Using gain- and loss-of-function approaches in vivo and in vitro, we proved that Tisp40 deficiency significantly exacerbated I/R-induced oxidative stress, apoptosis and acute cardiac injury, and facilitated cardiac remodeling and dysfunction following long-term observations, conversely, I/R-induced cardiac injury and dysfunction were ameliorated in Tisp40 cTG mice. To determine the molecular basis through which Tisp40 ameliorates cardiac I/R injury, we re-analyzed the microarray data from human prostate cells expressing a conditionally active Tisp40 (accession no. GSE7223). KEGG analysis revealed that the HBP was dramatically activated by Tisp40, accompanied by an increased N-glycan biosynthesis. Therefore, we then investigated whether HBP and protein O-GlcNAcylation were responsible for the cardioprotection of

Tisp40. In summary, the downstream targets of Tisp40 was screened by the unbiased microarray data.

As for the INTRODUCTION part, the manuscript was drafted after all the study being finished. To provide more background information to the reader, we introduced some previous researches about HBP, protein O-GlcNAcylation and Tisp40. Of note, no studies about the relationship between Tisp40 and O-GlcNAcylation are currently available, so we cannot link Tisp40 with O-GlcNAcylation.

Specific:

1. Lines 993–994 do not apply to Figure 8.

Response: Thanks for your comment. As suggested, we removed the sentences.

Reviewer #2 (Remarks to the Author):

In this study, Dr. Tang and colleagues aimed to characterize the role of Tisp40, an understudied protein in ER stress, cardio-protection, and hexosamine biosynthesis. The authors identified Tisp40 as an increased transcription factor by myocardial ischemia/reperfusion (I/R). In vivo tests demonstrated that Tisp40 protected the heart against I/R injury. At the mechanistic level, Tisp40 targeted GFPT1, the rate-limiting enzyme of hexosamine biosynthesis. Additional loss-of-function experiments revealed that GFPT1 mediated, at least partially, the protective effect of Tisp40 under I/R. Overall, the study was thoughtfully designed, and the conclusions are largely supported by the findings.

1. Figure 2, the IA in the global KO mice is quite high. I suspect that there will be mortality issue. Can the authors provide survival rate comparing WT and KO mice after I/R?

Response: Thanks for your insightful comment. There was also no significant difference in survival during 24 h of follow-up after cardiac IR injury (8.0% in WT mice vs. 12.0% in KO mice, $P > 0.05$). The data were provided in the last sentence of the first paragraph in revised “Tisp40 deficiency exacerbates oxidative stress, apoptosis and cardiac I/R injury in vivo and in vitro”.

2. Figure 2D, Figure 4B, Figure 4E, since there is no co-staining with cardiomyocyte marker, it is important to emphasize that the positive signal may not be cardiomyocytes. For example, they might be immune cells.

Response: Thanks for your insightful comment. To specially identify the apoptosis of cardiomyocytes in murine hearts, immunofluorescence staining with an anti- α -actinin antibody was performed to label cardiomyocytes. Apoptotic index was expressed as the percentage of apoptotic α -actinin positive cardiomyocytes over the total cardiomyocytes^[1-3]. The updated data were provided in revised Figure 2D, Figure 4B and related figures. As for the DHE staining in Figure 4E, DHE probe is widely used for examining superoxide anion in the myocardium; however, the signal is easily quenched. To detect superoxide anion in murine hearts accurately, we used fresh frozen cardiac slices, and the study was also quickly performed as previously described^[4-6]. Based on this reason, we cannot simultaneously identify cardiomyocytes with α -actinin co-staining. Meanwhile, we evaluated the effect of Tisp40 on free radicals generations using primary cardiomyocytes in vitro (Figure S4G-H and Figure S9I-J).

[1] Zhao YC, et al. Disruption of circadian rhythms by shift work exacerbates reperfusion injury in myocardial infarction. *J Am Coll Cardiol.* 2022; 79(21): 2097-2115.

[2] Matsuda T, et al. NF2 activates Hippo signaling and promotes ischemia/reperfusion injury in the heart.

Circ Res. 2016; 119(5): 596-606.

- [3] Woodall MC, et al. Cardiac fibroblast GRK2 deletion enhances contractility and remodeling following ischemia/reperfusion injury. *Circ Res.* 2016; 119(10): 1116-1127.
- [4] Yu L, et al. Megakaryocytic Leukemia 1 bridges epigenetic activation of NADPH oxidase in macrophages to cardiac ischemia-reperfusion injury. *Circulation.* 2018; 138(24): 2820-2836.
- [5] Matsushima S, et al. Broad suppression of NADPH oxidase activity exacerbates ischemia/reperfusion injury through inadvertent downregulation of hypoxia-inducible factor-1 α and upregulation of peroxisome proliferator-activated receptor- α . *Circ Res.* 2013; 112(8): 1135-1149.
- [6] Zhang X, et al. FNDC5 alleviates oxidative stress and cardiomyocyte apoptosis in doxorubicin-induced cardiotoxicity via activating AKT. *Cell Death Differ.* 2020; 27(2): 540-555.

3. The Y axis of all data should start from 0, not a random number. This issue is pertinent to Figures 3E, 3F, 3B, 4K, 5K, 6I, 7G, 7K, 7M, 7N, S2C, S6G, S6H, S7G, S8K, S9J, S12G, S15A, and S15H.

Response: Thanks for your comment. As suggested, we revised the figures.

4. It is interesting to know if knockout of Tisp40 affects the expression of other ER stress signaling transducers, such as IRE1a, XBP1s, ATF6, PERK, ATF4, and GRP78.

Response: Thanks for your insightful comment. As suggested, we examined the expression of other ER stress signaling transducers in Tisp40 KO hearts at 4 h and 8 h after cardiac I/R surgery^[1]. Yet, no alterations of IRE1a, XBP1s, ATF6, PERK, ATF4, and GRP78 expression were observed (shown below). Considering the fact that Tisp40 expression was induced by ER stressors (Figure 8), we proposed that Tisp40 might work downstream of ER stress. In addition, findings from Jin et al. also revealed that ATF6 overexpression did not affect the other 2 branches of the ER stress response^[2]. The results indicate that these ER stress-associated proteins may work in independent ways to restore ER homeostasis.

- [1] Wang ZV, et al. Spliced X-box binding protein 1 couples the unfolded protein response to hexosamine biosynthetic pathway. *Cell*. 2014; 156(6): 1179-1192.
- [2] Jin JK, et al. ATF6 decreases myocardial ischemia/reperfusion damage and links ER stress and oxidative stress signaling pathways in the heart. *Circ Res*. 2017; 120(5): 862-875.

5. Throughout the manuscript, the authors used H&E staining images to quantify cardiomyocyte size. This is not accurate. WGA staining is preferred. Moreover, the N number refers to the number of mice used or the number of cardiomyocytes? It is not clear. Along the same line, in addition to fibrosis as quantified by the authors, cardiac pathological remodeling can be assessed by gene expression of the fetal gene program, including NPPA, NPPB, beta-MHC, etc.

Response: Thanks for your insightful comment. As suggested, we re-evaluated cardiomyocyte size with WGA staining, and all relative data were re-quantified accordingly. The N number refers to the number of mice, not the number of cardiomyocytes. To avoid mis-understanding, we revised the description in methods "Histological analysis". Indeed, cardiac remodeling can be assessed by gene expression of the fetal gene program. The data (*Anp*, *α-Mhc*, *β-Mhc*, *Col1a1*, *Col3a1*, *Tgf-β1* and *Ctgf*) were already provided in Supplementary Figure 6A, Supplementary Figure 6D, Supplementary Figure 8F and Supplementary Figure 8I.

6. Does Tisp40 affect the transcription of other enzymes of the hexosamine biosynthetic pathway, such as GNPAT1, UAP1, etc?

Response: Thanks for your insightful comment. As suggested, we measured the mRNA levels of *Gnpnat1*, *Pgm3*, *Uap1*, *Ogt* and *Oga*, the key enzymes of HBP downstream of GFPT1, in Tisp40 cTG hearts as well as primary cardiomyocytes with Tisp40 overexpression. Interestingly, the mRNA levels of glucosamine-phosphate N-acetyltransferase (*Gnpnat1*) and phosphoglucomutase 3 (*Pgm3*), instead of UDP-N-acetylglucosamine pyrophosphorylase 1 (*Uap1*), *Ogt* or *Oga*, were significantly increased in Tisp40 cTG hearts at baseline. Moreover, overexpression of an active Tisp40 in NRCMs also induced the expression of *Gnpnat1* and *Pgm3* at baseline. These data were consistent with previous studies identifying a conserved UPRE DNA motif in GNPAT1 and PGM3 promoters^[1,2]. The data were provided as Supplementary Figure 17A-B, and the primers were listed in Supplementary Table 2.

[1] Wang ZV, et al. Spliced X-box binding protein 1 couples the unfolded protein response to hexosamine biosynthetic pathway. *Cell*. 2014; 156(6): 1179-1192.

[2] Deng Y, et al. The Xbp1s/GaIE axis links ER stress to postprandial hepatic metabolism. *J Clin Invest*. 2013; 123(1): 455-468.

Reviewer #3 (Remarks to the Author):

Zhang et al TSP40 NCOMMS-23-00930 " Tisp40, a novel cardiomyocyte-enriched UPR-associated transcription factor, prevents cardiac ischemia/reperfusion injury through the hexosamine biosynthetic pathway ". This manuscript describes studies done mostly in a mouse model of ischemia/reperfusion injury (IRI) and testing how Tisp40, also known as Creb311, affects reperfusion injury. The study is clearly described and the results are well depicted and appropriately interpreted.

Comments:

1-The authors should refrain from concluding that I/R or ER stress causes the cleavage of the ER form of Tisp40 and the translocation of the product to the nucleus. They showed that there are two forms of Tisp40, one larger than the other, and that the smaller of the forms concentrates in the nucleus after stress, but this does not prove their cleavage/translocation theory.

Response: Thanks for your insightful comment. Tisp40 is a well-known endoplasmic reticulum membrane (ER)-resident type II transmembrane protein, anchored to the ER through its transmembrane domain, and then released to the nucleus by a “regulated intramembrane proteolysis (Rip)” cleavage mechanism (sequentially cleaved by S1P and S2P), where it functions as an active transcription factor^[1, 2]. In addition, Nagamori et al. have found that Tisp40 protein contains the S1P and S2P recognition motifs RXXL and LXXXLXXXP (also shown in Figure 1A of our study). By co-transfection of Tisp40 with S1P, they demonstrated that Tisp40 was processed in the Rip pathway^[2]. In our study, Tisp40 was found to be weakly expressed in the cytoplasm (especially in the ER) of cardiomyocytes in vivo and in vitro at baseline, which was then upregulated and processed to a smaller fragment by ER stress in I/R-injured hearts or cardiomyocytes. Using western blot and immunofluorescence staining, we confirmed this smaller fragment was the nuclear Tisp40. Based on the information of Tisp40 and the previous reports about UPR-associated transcription factors (ATF6, etc), we reasonably concluded that Tisp40 expression, cleavage and nuclear translocation were induced by ER stress under I/R injury. To make the results easily to read, we put some key background information and references in RESULTS “3.1 Tisp40 expression and nuclear translocation are induced by cardiac I/R injury”. We hope our explanation makes you satisfactory.

[1] Stirling J, et al. CREB4, a transmembrane bZip transcription factor and potential new substrate for regulation and cleavage by S1P. *Mol Biol Cell*. 2006; 17(1): 413-426.

[2] Nagamori I, et al. Tisp40, a spermatid specific bZip transcription factor, functions by binding to the unfolded protein response element via the Rip pathway. *Genes Cells*. 2005; 10(6): 575-594.

2-Related to my previous comment, it is premature to show a conclusion diagram such as in Fig. 8F, since much of the processes shown are hypothetical. This panel should be removed.

Response: Thanks for your thoughtful comment. This comment was addressed in the front one. To make the conclusion diagram more accurate, we labelled the translocation mechanism with a dotted line.

3-The Tisp40 knock out mice appear to have this gene deleted globally. A more suitable model would have been to use cardiac specific knock out, which the authors did use later for ectopic expression of Tisp40. However, given all of the other evidence the authors have provided, this is potential excusable. The authors should say in the Results that the Tisp40 knock out is global.

Response: Thanks for your comment and thoughtful consideration. As you indicated, we used a global Tisp40 knockout mice in the loss-of-function studies. To address the cardiac-specific role of Tisp40, we generated a cardiac-specific Tisp40 transgenic mice in the gain-of-function study. Meanwhile, we overexpressed the active Tisp40 (AAV9 Δ TM-HA) in the myocardium from Tisp40 KO mice to evaluate the cell autonomous effect in the myocardium. Besides, we also performed bone marrow transplantation to exclude the role of myeloid Tisp40 in cardiac I/R injury using global Tisp40 KO mice. Moreover, the studies in primary cardiomyocytes also confirmed the role of Tisp40 in cardiomyocytes. As suggested, we stated that we used global Tisp40 KO mice in “Tisp40 deficiency exacerbates oxidative stress, apoptosis and cardiac I/R injury in vivo and in vitro”, and the sentence was labelled with YELLOW. In addition, we also addressed this mice in other parts of the manuscript and labelled them with YELLOW.

4-The authors should remove the word “novel” wherever it appears in the manuscript. Tisp40 is not novel, it has been previously studied, albeit in other cell/tissue types.

Response: Thanks for your comment. As suggested, we removed the word “novel”.

REVIEWERS' COMMENTS

Reviewer #1 (Remarks to the Author):

No additional remarks.

Reviewer #2 (Remarks to the Author):

The authors have addressed mu previous concerns.

Reviewer #3 (Remarks to the Author):

The authors have sufficiently addressed my previous comments on their manuscript.

Reviewer #1 (Remarks to the Author):

No additional remarks.

Response: Thanks for your effort and consideration on our manuscript.

Reviewer #2 (Remarks to the Author):

The authors have addressed mu previous concerns.

Response: Thanks for your effort and consideration on our manuscript.

Reviewer #3 (Remarks to the Author):

The authors have sufficiently addressed my previous comments on their manuscript.

Response: Thanks for your effort and consideration on our manuscript.